# Automated Creativity Evaluation for LLMs with Semantic Entropy and Efficient Multi-Agent Judging Across Open-Ended Tasks

## Abstract

Large language models (LLMs) have achieved remarkable progress in language understanding, reasoning, and generation, sparking growing interest in their creative potential. Realizing this potential requires systematic and scalable methods for evaluating creativity across diverse tasks. However, most existing creativity metrics are tightly coupled to specific tasks, embedding domain assumptions into the evaluation process and limiting scalability and generality. To address this gap, we introduce an automated, domain-agnostic framework for quantifying LLM creativity across open-ended tasks. Our approach separates the measurement apparatus from the creative task itself, enabling scalable, task-agnostic assessment. Divergent creativity is measured using semantic entropy—a reference-free, robust metric for novelty and diversity, validated against LLM-based novelty judgments and baseline diversity measures. Convergent creativity is assessed via a novel retrieval-based multi-agent judge framework that delivers context-sensitive evaluation of task fulfilment with over 60% improved efficiency. We validate our framework across three qualitatively distinct domains—problem-solving (Mac-Gyver), research ideation (HypoGen), and creative writing (BookMIA)—using a broad suite of LLMs. Empirical results show our metrics reliably capture key facets of creativity—novelty, diversity, and task fulfilment—and reveal how model properties such as size, temperature, recency, and reasoning impact creative performance. Our work establishes a reproducible, generalizable standard for automated LLM creativity evaluation, paving the way for scalable benchmarking and accelerating progress in creative AI.

## 1 Introduction

Recent advances in large language models (LLMs) have led to major breakthroughs in language comprehension, generation, and reasoning (Lewis et al., 2019; Manning, 2022; Cobbe et al., 2021). As LLMs become more adept at reasoning and planning, their creative potential has emerged as a key area of interest (Ye et al., 2024; Sun et al., 2024). Creative LLMs can accelerate scientific discovery by proposing unconventional solutions (Ruan et al., 2024; Gu et al., 2024), uncovering novel patterns (Si et al., 2024), and automating experiment design (Liu et al., 2024), with far-reaching applications in materials science (Centre), research methodology (Boyko et al., 2023), and causal discovery (Li et al., 2025). Understanding and quantifying these creative capabilities is thus increasingly important.

However, most existing creativity evaluation frameworks are tightly coupled to specific tasks or domains, embedding strong domain assumptions into the assessment process (Krašovec, 2024; Kroll & Kraus, 2024). These approaches rely on curated answer sets, hand-crafted rubrics, or extensive human annotation—rendering creativity assessment subjective, resource-intensive, and difficult to scale, and leaving the field without automated, domain-general evaluation standards.

To address these challenges, we propose a fully automated, domain-general framework for evaluating LLM creativity that is both robust and scalable across open-ended tasks. Our framework decouples evaluation from specific creative tasks, enabling systematic, reference-free assessment of model creativity across domains. Building on cognitive science, which characterizes creativity as

encompassing both divergent and convergent thinking (Guilford, 1950), we deliberately design our framework to evaluate both aspects through novel, automated methods.

Divergent thinking is the ability to generate diverse, novel, and innovative ideas. We argue that significant semantic differences in LLM outputs can reflect divergent thinking by producing unconventional ideas. To capture this, we adapt *Semantic Entropy*, a sampling-based, reference-free metric quantifying the variability of model-generated outputs. We further validate its utility by benchmarking semantic entropy against LLM-based novelty judgments and additional diversity baselines, finding that it faithfully reflects core markers of divergent creativity.

Convergent thinking involves synthesizing information to produce solutions tailored to specific goals and contexts (Kumar et al., 2024). Recognizing the inherent subjectivity in evaluating this aspect (Li et al., 2023), we propose an adaptable, autonomous multi-agent LLM judging framework, where agents collaboratively assess distinct facets of task fulfilment (Lu et al., 2024a). To address the computational inefficiency of traditional discussion-based evaluations (Wang et al., 2024a), we introduce a retrieval-based discussion framework that streamlines the review process, making large-scale benchmarking more feasible.

To demonstrate the generality and practical value of our framework, we evaluate it across three qualitatively distinct domains of creativity: **problem-solving** with the MacGyver dataset (Tian et al., 2024), **research ideation** with the HypoGen dataset (O'Neill et al., 2025), and **artistic creativity** through creative writing with the BookMIA dataset (Shi et al., 2024). Together, these domains capture complementary facets of creativity—functional reasoning under constraints, logical synthesis of scientific ideas, and narrative originality. We apply both methods to 300 problems per domain and benchmark diverse LLMs, analyzing how model size, recency, temperature, and reasoning augmentation affect creativity.

In summary, we: **(1)** introduce a reference-free, automated assessment of divergent creativity based on semantic entropy; **(2)** develop a more compute-efficient multi-agent LLM judging framework for convergent creativity; and **(3)** provide comprehensive empirical benchmarking of LLMs' creativity across MacGyver, HypoGen, and BookMIA—together establishing an automated, domain-general LLM creativity evaluation framework. Our experiments show that semantic entropy reliably captures creative breadth and diversity, our multi-agent framework achieves human-level accuracy with substantial efficiency gains, the framework generalizes across distinct domains of creativity, and we further investigate how model size, recency, and reasoning parameters affect creative performance.

## 2 RELATED WORK

**Human Creativity Tests.** Classic human creativity assessments—such as Torrance Tests of Creative Thinking (TTCT) and Consensual Assessment Technique (CAT) (Torrance; Amabile, 1982)—have been adapted to evaluate LLMs. However, these methods depend on extensive human annotation, making them unscalable and ill-suited for automated evaluation. Moreover, while TTCT metrics like fluency and elaboration are meaningful in human settings, they are less reliable for LLMs, since idea count and output length can be trivially adjusted by sampling. Consequently, our framework focuses on originality and flexibility (divergent creativity), which remain robust indicators for LLM generation tasks, and utilizes a separate, automated judge for task fulfilment (convergent creativity).

**Domain-specific Creativity Evaluation.** Beyond classic human tests, a wide range of task-specific creativity benchmarks have been developed for LLMs, spanning mathematical reasoning, hardware design, metaphor generation and code synthesis (Ye et al., 2024; DeLorenzo et al., 2024; Paul V. DiStefano & Beaty, 2024; Gómez-Rodríguez & Williams, 2023). These frameworks typically embed strong domain assumptions, require curated answer sets or subjective metrics, and are closely tied to the structure of their target tasks. Hence, they lack generalizability and are difficult to apply systematically to the open-ended challenges tackled by LLMs. Our framework overcomes these limitations by providing a task-agnostic, reference-free, and fully automated approach to creativity evaluation.

**Divergent Creativity Evaluation.** Automated metrics for divergent creativity in LLMs—such as semantic similarity, integration scores, and Lempel–Ziv complexity—offer some insight into output diversity, but often miss the nuance required for complex, open-ended tasks (Mohammadi, 2024; Chen & Ding, 2023; Summers-Stay et al., 2023; Peeperkorn et al., 2024; Bellemare-Pepin et al., 2024). Recent work has instead used uncertainty to detect hallucinations in LLM outputs (Huang

et al., 2024; Chen et al., 2025; Zhang et al., 2023; Sriramanan et al., 2024), including one based on Semantic Entropy (SE)(Farquhar et al., 2024). We build on this by repurposing Semantic Entropy, originally for hallucination detection, as a robust, reference-free measure of divergent creativity.

**Convergent Creativity Evaluation.** Convergent creativity is the ability to refine, select, and deliver solutions that are useful, feasible, and context-appropriate. Foundational creativity research (Torrance; Guilford, 1950; Cropley, 2006; Amabile, 1983; Runco & Jaeger, 2012) consistently defines creativity as producing ideas that are both novel (divergent) and useful or appropriate (convergent). While divergent creativity captures the generation of original and varied ideas, convergent creativity evaluates whether those ideas effectively solve the problem, satisfy constraints, and are practically viable. This distinction is critical, since models may produce diverse outputs that are incoherent or irrelevant; without convergent evaluation, such outputs would misleadingly appear "creative."

Traditional tests like the Remote Associates Test (RAT) (Mednick & Mednick, 1967) are not well suited to LLMs, as they were designed for humans. Recent pipelines leverage LLMs as judges (Rabeyah et al., 2024; Dubois et al., 2024; Li et al., 2024; Zheng et al., 2023), with multi-agent discussion frameworks shown to provide more nuanced and comprehensive evaluation of candidate solutions (Liang et al., 2024; Chan et al., 2023). However, these methods are computationally intensive and hard to scale (Lv et al., 2024; Luo et al., 2023). Our novel retrieval-based discussion framework addresses this, enabling scalable, robust evaluation of task fulfilment.

## 3 DIVERGENT CREATIVITY

### 3.1 BACKGROUND ON SEMANTIC ENTROPY

**Semantic Clustering.** Following Farquhar et al. (2024), Step generations $(s_1...s_n)$ are clustered using bi-directional entailment, where a greedy algorithm assigns each generation to an existing class $C_a$ if sufficiently similar, or creates a new class otherwise.

**Semantic Entropy.** For a query $x$, the probability, $P(s|x)$, of a generated step $s$, comprising tokens $(t_1, ..., t_i)$ is given by the product of its conditional token probabilities. For computational efficiency, we use log-probability $\log P(s|x)$.

$$\log P(s|x) = \sum_i \log P(t_i|t_{<i}, x) \tag{1}$$

The probability of a semantic class $c$ is the sum of all generated samples $s$ belonging to the class:

$$P(c|x) = \sum_{s \in c} P(s|x) \tag{2}$$

Semantic Entropy is computed as the entropy of the class probability distribution over all classes $C$:

$$H(x) = -\sum_{i=1}^{|C|} P(C_i|x) \log P(C_i|x) \tag{3}$$

### 3.2 AUTOMATED DIVERGENT CREATIVITY EVALUATION WITH SEMANTIC ENTROPY

Hallucination-like processes in humans reflect associative thinking, a key mechanism underlying creativity (Jiang et al., 2024; Raffaelli et al., 2024; Ritter & Dijksterhuis, 2014). By making unexpected connections, associative thinking enables the generation of multiple, varied, or unconventional ideas—a hallmark of divergent thinking (Guilford, 1950). We hypothesize that, in LLMs, generation uncertainty—where the model produces unpredictable or surprising outputs—similarly signals divergent creativity by reflecting an ability to explore novel solution paths.

**Motivation.** To robustly quantify this breadth and novelty in model outputs, we use Semantic Entropy (Farquhar et al., 2024), which measures the unpredictability and diversity of generated solutions at the semantic level. Unlike word-level entropy or surface-level diversity metrics, semantic entropy captures true conceptual differences, identifying outputs that are novel in substance rather than just rephrasings. This reference-free and scalable approach enables automated creativity assess-

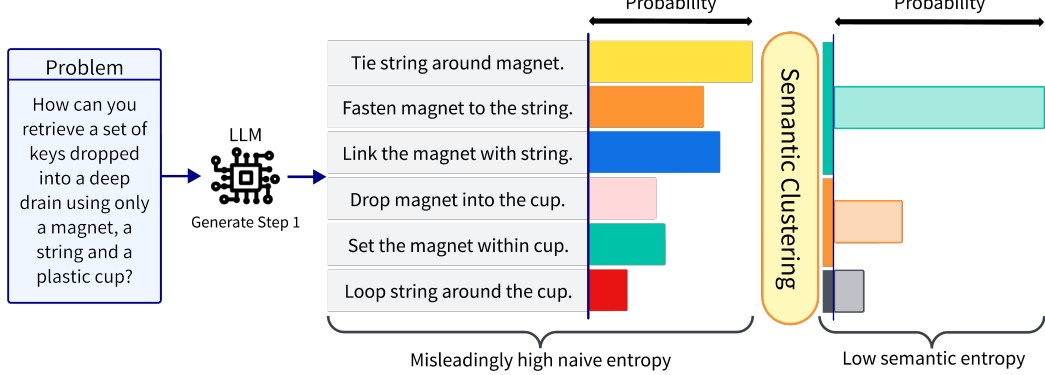

Figure 1: Semantic Entropy: LLM-generated steps clustered by similarity, with entropy computed over cluster probabilities. Naive entropy (middle) uses raw probabilities; Semantic Entropy (right) clusters by meaning for a more reliable measure.

ment across domains. Because unpredictability and diversity are closely linked to creativity markers like originality and flexibility, we will further investigate how semantic entropy aligns with established creativity metrics, leveraging both LLM-based novelty judgments and diversity baselines.

**Implementation.** We generate solutions step by step: at each stage, we sample $n = 10$ candidate solutions per step, cluster them by semantic equivalence, and compute Semantic Entropy over the resulting class probabilities as seen in Figure 1. The highest-probability sample is iteratively appended to build a full solution, repeating until majority of the samples indicate completion ("STOP").

**Entailment Model.** Our semantic clustering procedure requires an entailment model to group generated samples into semantic classes by assessing equivalence. Importantly, the benchmark itself is model-agnostic: the clustering algorithm is compatible with any Natural Language Inference (NLI) model, and researchers are free to substitute their preferred encoder. We use `tasksource/deberta-base-long-nli` in our implementation, because it achieved the highest accuracy (Table 1) on human-annotated sentence pairs from the Macgyver dataset, which we considered essential to ensure the reliability and consistency of downstream analyses. See Appendix C.2 for full entailment and ground-truth procedure.

Table 1: Entailment models.

| Model | Accuracy |
|---|---|
| DeBERTa-Base-Long-NLI | **78.1%** |
| GPT-4o | 72.7% |
| Bart-Large-MNLI | 52.7% |
| DeBERTa-v3-Large | 52.7% |
| RoBERTa-Large-MNLI | 47.2% |
| DeBERTa-v3-Large (GLUE MNLI) | 67.3% |

## 4 CONVERGENT CREATIVITY

**Motivation.** Evaluating convergent creativity in LLMs requires assessing how well generated solutions fulfil diverse goals and constraints across domains. LLM-as-a-judge frameworks are a widely adopted, robust approach for automating open-ended evaluation, enabling large-scale, accurate, consistent assessment while overcoming limitations of human annotation (Badshah & Sajjad, 2024; Gu et al., 2025). Unlike single-task or fixed-rubric benchmarks, our framework supports configurable, metric-specific evaluation that flexibly captures domain-specific criteria—essential for creative tasks where "success" is context-dependent. Our automated approach enables scalable, reference-free quantification of task fulfilment across a broad range of open-ended problems.

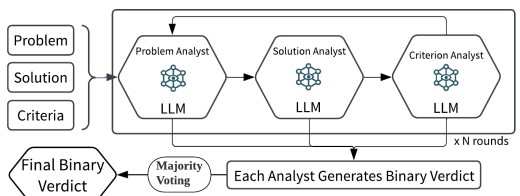

Figure 2: ChatEval (One-by-one) framework (Chan et al., 2023) and its information flow.

**Current challenges.** Multi-agent judge frameworks (Liang et al., 2024; Chan et al., 2023) such as ChatEval (Figure 2) yield more nuanced, context-aware assessments by having LLMs engage in discussion—outperforming one-shot or few-shot judging, as each agent can identify distinct aspects and subtleties in a solution. However, appending the full discussion history at each turn is highly resource-intensive, quickly inflating token usage and computation, and making it impractical for large-scale benchmarks.

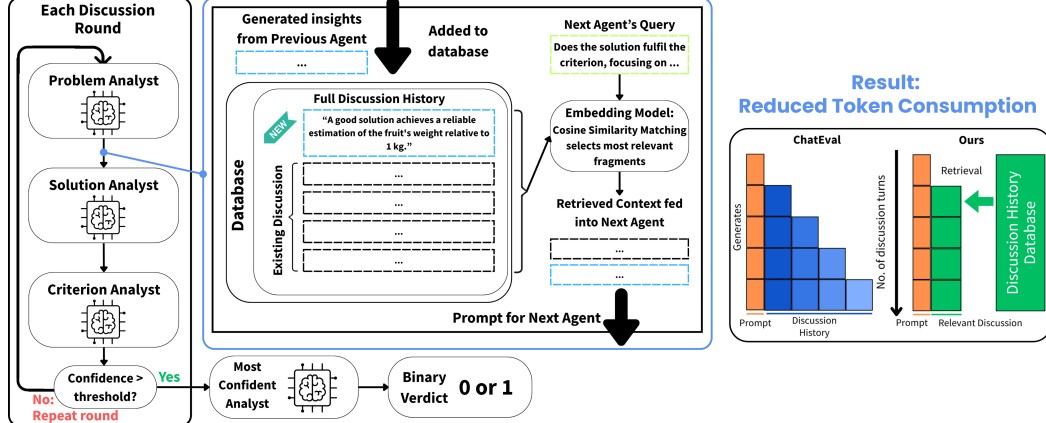

Figure 3: Retrieval-Based Multi-Agent Framework. **Left:** Three specialised LLM agents—Problem, Solution, and Criterion—analyze tasks from different perspectives, recording insights with confidence scores. **Middle:** Fragments are embedded in a vector database; each agent retrieves only the $k$ most relevant fragments via cosine similarity during their turn. **Right:** This retrieval loop cuts token usage by ≈63% compared to ChatEval while converging on final binary verdicts.

## 4.1 AUTOMATED CONVERGENT CREATIVITY EVALUATION

**Retrieval-based Framework.** To address the inefficiency of standard multi-agent LLM judging, we introduce a retrieval-based framework (Fig. 3) that preserves nuanced evaluation while drastically reducing token usage (by 63% vs. ChatEval; see Appendix D.4). Agents attend only to the most relevant prior discussion fragments, with early stopping and confidence scoring further limiting redundant deliberation, making scalable, context-aware assessment feasible across diverse datasets.

**Implementation Overview.** As illustrated in Fig. 3, our framework structures discussion among three specialized LLM agents—Problem, Solution, and Criterion analysts—that collaboratively evaluate solutions from different perspectives. Each agent contributes insights as retrievable fragments, which are iteratively expanded and synthesized into a final binary verdict. Full prompt templates, mathematical formalization, and implementation details are provided in Appendix D.2.

## 5 EXPERIMENTAL SETUP

**Domains and Evaluation Metrics.** Creativity spans a wide range of human activities and intellectual pursuits. Following the broad taxonomy introduced by Ismayilzada et al. (2025), we consider three complementary domains: unconventional problem-solving, research ideation, and artistic creativity. In our framework, these are instantiated through three representative datasets: **MacGyver** (problem-solving) (Tian et al., 2024), **HypoGen** (research ideation) (O'Neill et al., 2025), and **Book-MIA** (creative writing as artistic creativity) (Shi et al., 2024). Each domain emphasizes a distinct dimension of creativity—practical reasoning under constraints, synthesis of novel scientific hypotheses, and narrative originality—together providing a diverse and balanced testbed for assessing the domain-generality of our framework. We describe each dataset and its evaluation metrics below.

- **MacGyver:** Real-world physical reasoning problems requiring creative, unconventional use of everyday objects. Models are prompted to generate step-by-step solutions.
  Metrics: **Feasibility, Safety, Effectiveness**

- **Hypogen:** Open-ended scientific ideation tasks(O'Neill et al., 2025). Each problem presents a standard "Bit" and a target "Flip"(see Fig. 4); models must generate a novel chain of reasoning from Bit to Flip without being shown the ground-truth explanation.
  Metrics: **Feasibility, Scientific Accuracy, Relevance**

- **BookMIA:** Creative writing tasks(Shi et al., 2024), where models are challenged to generate multi-paragraph narratives that logically connect provided starter and ending sentences, mirroring real-world narrative composition (Yi et al., 2025).
  Metrics: **Coherence, Emotional and Psychological Realism, Plot Completion**

*See appendix D.1 for sample prompts, generations and full metric definitions.*

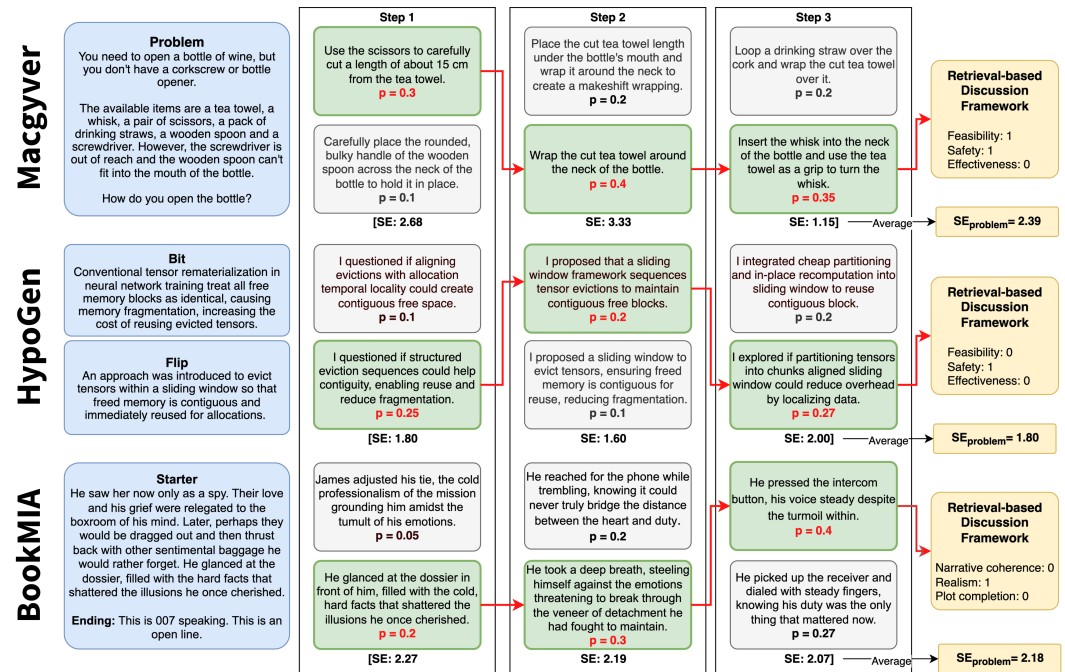

Figure 4: Overview of our benchmark, for all 3 datasets.

**Divergent Creativity Verification.** We benchmark Semantic Entropy against relative novelty and diversity baselines. For novelty, a pairwise LLM judge ranks solution originality for 50 MacGyver and BookMIA problems per model; reliability was validated against the average rankings from five human annotators, each ranking 30 solutions (Pearson correlation of 0.80). Pipeline and annotation details are in the appendix C.6. For diversity, we investigate average cosine similarity between clusters against Semantic Entropy, with other metrics (e.g., self-BLEU) in the appendix C.3.

**Convergent Creativity Verification.** To assess the reliability of our automated Multi-Agent Judge framework, we compare its verdicts to a "golden truth" obtained by majority vote from five human annotators on a randomly sampled set of 50 problems on the Macgyver dataset. We report accuracy as Accuracy-Rejection Curve (AUARC)(Nadeem et al., 2009), and detail the annotation protocol and inter-annotator agreement in the appendix.

**Detailed Pipeline.** We benchmark LLMs on the MacGyver, HypoGen and BookMIA datasets using our unified framework. For each model, 300 problems per domain are solved step by step: at each stage, 10 candidate next steps are generated and clustered to compute semantic entropy, then the most likely candidate is selected (greedy search) and appended to the solution. Divergent creativity is reported as the average entropy across all steps in the solution, reflecting overall exploration.

Convergent creativity is assessed on all 300 generated solutions using our Multi-Agent Judge and domain-specific metrics. Final model scores include both Divergent (Semantic Entropy) and Convergent (Multi-Agent Judge accuracy) results. All experiments used 4 NVIDIA A100 GPUs, with API access for large models. Further details are in the appendix.

## 6 RESULTS AND DISCUSSION

### 6.1 DIVERGENT CREATIVITY

**Semantic entropy robustly captures the breadth and flexibility of idea generation in LLMs.** We find that semantic entropy, computed over semantically distinct clusters, is strongly correlated with the number of unique idea categories (Fig. 5a), indicating that it reliably reflects flexibility—one of the four TTCT metrics. Semantic entropy is also highly responsive to temperature: as temperature increases, models generate more varied and less repetitive outputs, resulting in higher entropy across architectures (Fig. 5b), consistent with findings that increased temperature enhances diversity by reducing repetition and encouraging creative risk-taking (Chen & Ding, 2023; Roemmele & Gordon, 2018). Notably, while semantic entropy rises rapidly with temperature at first, it plateaus at higher values, likely reflecting saturation in generating meaningful, distinct outputs (Chen & Ding, 2023). This establishes semantic entropy as a robust and scalable metric for quantifying generative range and creative flexibility of LLM solutions—key markers of creative potential.

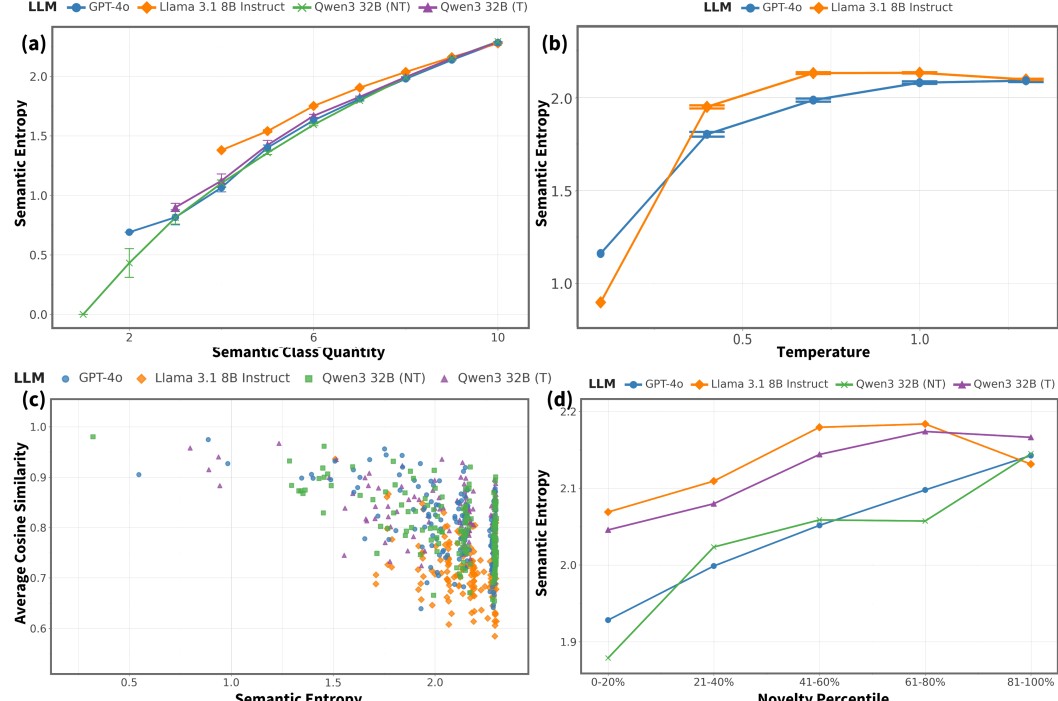

Figure 5: Semantic entropy's relationship with response and model parameters. In this and subsequent figures, (T) and (NT) refers to (Thinking) and (Non-thinking) respectively.

**Semantic entropy meaningfully tracks core creative attributes, aligning with both novelty and diversity baselines.** To further validate semantic entropy as a practical proxy for divergent creativity, we benchmarked it against an LLM-based pairwise novelty judge (validated with strong agreement to human annotators) and established diversity metrics, such as average cosine similarity between solutions. Our results show that semantic entropy is positively associated with both judged novelty (Fig. 5d) and lower cosine similarity (Fig. 5c), indicating that models with higher entropy not only produce more diverse ideas, but also outputs considered more original, corroborating previous work which observed correlations between novelty and predictive uncertainty (Chen et al., 2025). These findings reinforce semantic entropy's utility as an automated, reference-free, and domain-general indicator for core creative attributes—enabling principled evaluation of LLM creativity at scale.

**The advancement and size of LLMs does not correlate with divergent creativity.** As shown in the appendix H, semantic entropy remains largely stable—and sometimes even decreases—as models become larger or newer, both across Llama generations (3, 3.1, 3.3; 8B to 405B) and Vicuna model sizes (7B to 33B). This is likely due to training that prioritises convergent solutions (Yu et al., 2024), potentially limiting divergent output in larger models and suggesting a developmental trajectory for creativity that is distinct from general advances in problem-solving or reasoning. Notably, Ruan et al. (2024) also found that less advanced and state-of-the-art models generate comparable levels of creative ideas in scientific contexts.

## 6.2 Convergent Creativity

**Larger, more recent and reasoning LLMs achieve higher task fulfilment.** Larger and more recent models like GPT-4o and Llama 3.1 70B consistently outperform earlier models such as GPT-3.5 and Llama 3.1 8B on convergent creativity metrics (Fig. 6a, 6b), which reflect a model's ability to generate solutions that meet explicit task requirements. This pattern is consistent with prior work demonstrating GPT-4o's advantages in code generation (Lu et al., 2024b), reasoning (Minaee et al., 2024), and Llama 70B's edge in instruction following (Kovalevskyi, 2024), largely attributed to scaling laws and advanced training strategies like instruction tuning and dataset diversification (Zhao et al., 2024). Reasoning-focused models such as R1-70B also outperform their non-reasoning base versions (e.g., Llama 3.3, Fig. 6c), reinforcing the value of reasoning for enhancing LLMs' abilities to tackle complex, multi-criteria tasks (DeepSeek-AI et al., 2025; Huang & Chang, 2023). Overall, these results show that scaling, recency, and improved reasoning directly boost LLMs' capacity for task fulfilment in automated convergent creativity evaluation.

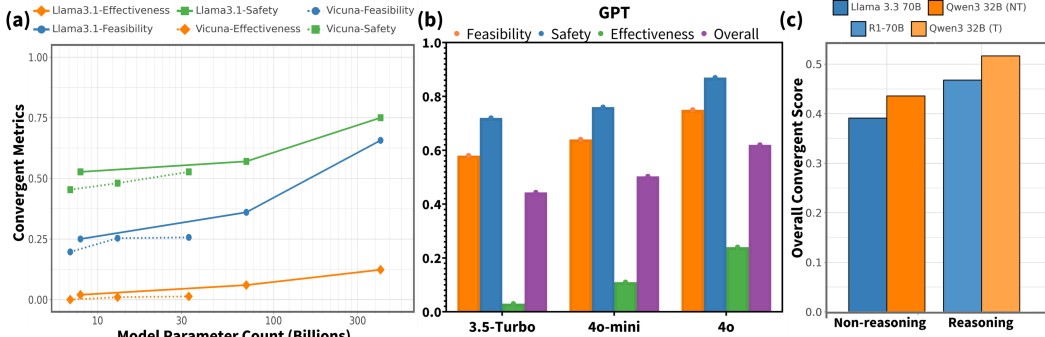

Figure 6: The impact of various parameters (left: model size, center: model recency, right: reasoning capabilities) on the convergent creativity of LLMs on the MacGyver dataset.

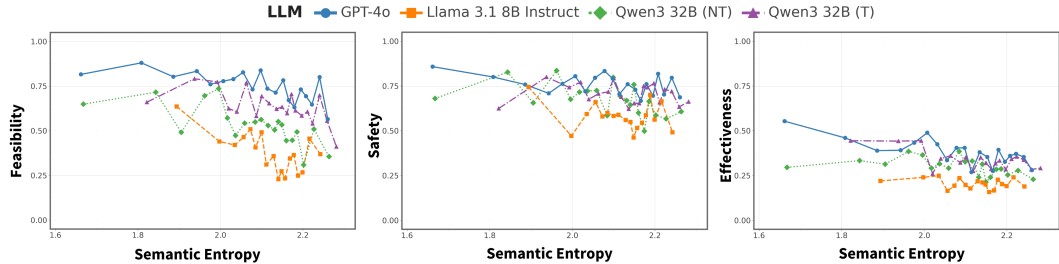

Figure 7: Semantic Entropy compared to different convergent creativity metrics (Y-axis) on the MacGyver dataset. Each point represents the mean Y value at the median X value of a unique set of 15 data points (fixed-interval binning). Similar trends were observed on the Hypogen and BookMIA datasets (see Appendix 15).

**Divergent and Convergent creative ability in LLMs arise from distinct mechanisms.** The relationship between divergent and convergent creativity is model-dependent, not strictly antagonistic. As shown in Figure 7, some LLMs (like GPT-4o) show a mild trade-off—higher semantic entropy can correspond to lower task fulfilment—while others (like Llama 8B) maintain more balanced performance. Our findings suggest that the mechanisms supporting divergent and convergent creativity can be at least partially decoupled, with the potential for both to be improved in tandem. Thus, maximizing creative potential does not necessarily require sacrificing convergent abilities. Further correlation results between divergent and convergent creativity are in appendix G.

**Our retrieval-based multi-agent judge enables robust, scalable assessment of convergent creativity across diverse domains.** We demonstrate that our retrieval-based multi-agent framework achieves accuracy and AUARC comparable to individual human annotators (see Table 2), and consistently outperforms all single-agent and baseline LLM judging pipelines, highlighting its effectiveness in capturing nuanced, context-dependent aspects of task fulfilment. By leveraging retrieval and confidence-based stopping, our method reduces computational cost by over 60% (see Appendix 15) compared to traditional multi-agent discussions (Chan et al., 2023), making large-scale evaluation feasible. These results show that automated, discussion-based LLM judges can match or exceed the reliability of human raters, while enabling efficient, repeatable assessment of LLM performance in open-ended, multi-criteria tasks across domains.

| Framework | Accuracy | AUARC |
|---|---|---|
| **Baselines** | | |
| One-shot | 64.7% | 0.693 |
| CoT | 67.3% | 0.697 |
| Few-shot | 65.3% | 0.720 |
| Few-shot w/CoT | 66.0% | 0.725 |
| ChatEval | 76.7% | - |
| **Our framework** | | |
| GPT-4o-mini | 55.3% | 0.635 |
| GPT-4o | **84.7%** | **0.907** |
| **Human** | | |
| Annotator1 | 82.7% | - |
| Annotator2 | **84.7%** | - |
| Annotator3 | 81.3% | - |
| Annotator4 | 80.0% | - |
| Annotator5 | 81.3% | - |

Table 2: Performance of different evaluation frameworks and human annotators, judging 50 solutions to Macgyver Dataset.

Apart from the findings above, we also analysed the effect of: (1) temperature on convergent creativity, (2) sample size on semantic entropy, (3) effect of step number on semantic entropy and (4) varying confidence thresholds on our framework's accuracy. The detailed analyses are in appendix H. Other ablation studies for semantic entropy, including comparisons to naive entropy and alternative step-aggregation methods(min, max) have been conducted in C.7.

Table 3: Performance of various LLMs on our benchmark using the MacGyver dataset.

| Model | Divergent Creativity | Convergent Creativity | | | |
|---|---|---|---|---|---|
| | Semantic Entropy | Feasibility | Safety | Effectiveness | Overall |
| Vicuna 7B | **2.19** | 0.20 | 0.45 | 0.00 | 0.22 |
| Vicuna 13B | 1.96 | 0.25 | 0.48 | 0.01 | 0.25 |
| Vicuna 33B | 2.17 | 0.26 | 0.53 | 0.01 | 0.26 |
| Llama 3 70B Instruct | 2.10 | 0.39 | 0.65 | 0.02 | 0.36 |
| Llama 3.1 8B Instruct | 2.13 | 0.25 | 0.53 | 0.02 | 0.27 |
| Llama 3.1 70B Nemotron Instruct | **2.19** | 0.36 | 0.57 | 0.06 | 0.33 |
| Llama 3.1 405B Instruct | 2.08 | 0.66 | 0.75 | 0.12 | 0.51 |
| Llama 3.3 70B Instruct | 2.10 | 0.45 | 0.68 | 0.04 | 0.39 |
| Deepseek R1 70B Distilled | 2.10 | 0.58 | 0.75 | 0.07 | 0.47 |
| GPT 3.5 Turbo | 2.02 | 0.51 | 0.71 | 0.03 | 0.42 |
| GPT 4o mini | 2.05 | 0.62 | 0.76 | 0.12 | 0.50 |
| GPT 4o | 2.08 | **0.82** | **0.86** | **0.21** | **0.63** |
| Qwen3 32B (Thinking) | 2.02 | 0.65 | 0.78 | 0.12 | 0.52 |
| Qwen3 32B (Non-think) | 2.08 | 0.49 | 0.74 | 0.08 | 0.44 |

Table 4: Performance of various LLMs on our benchmark using the HypoGen dataset.

| Model | Divergent Creativity | Convergent Creativity | | | |
|---|---|---|---|---|---|
| | Semantic Entropy | Feasibility | Relevance | Scientific Accuracy | Overall |
| GPT-4o | **2.07** | 0.28 | 0.61 | 0.17 | 0.35 |
| Llama 3.1 8B Instruct | 2.04 | 0.21 | 0.56 | 0.12 | 0.29 |
| Qwen3 32B (Thinking) | 1.72 | 0.41 | **0.78** | 0.21 | 0.47 |
| Qwen3 32B (Non-think) | 1.66 | **0.50** | 0.76 | **0.26** | **0.51** |

Table 5: Performance of various LLMs on our benchmark using the BookMIA dataset.

| Model | Divergent Creativity | Convergent Creativity | | | |
|---|---|---|---|---|---|
| | Semantic Entropy | Coherence | Realism | Plot Completion | Overall |
| GPT-4o | 2.17 | 0.36 | 0.40 | 0.23 | 0.33 |
| Llama 3.1 8B Instruct | 1.89 | 0.03 | 0.04 | 0.03 | 0.03 |
| Qwen3 32B (Thinking) | **2.19** | **0.50** | **0.41** | **0.52** | **0.48** |
| Qwen3 32B (Non-think) | **2.19** | 0.36 | 0.44 | 0.35 | 0.38 |

## 7 CONCLUSION

**Key findings and Broader Impact.** We introduce a fully automated, domain-general framework for evaluating LLM creativity, leveraging semantic entropy as a reference-free metric for divergent thinking and a retrieval-based multi-agent judge for convergent evaluation. Our experiments across the MacGyver, Hypogen and BookMIA benchmarks demonstrate several key findings: semantic entropy robustly quantifies the generative breadth and diversity of model outputs, correlating with established creativity markers like flexibility, novelty, and diversity. Our multi-agent judging framework achieves human-level reliability with over 60% improved computational efficiency, enabling practical, large-scale assessment of LLM task fulfilment in diverse, open-ended settings. In contrast to convergent performance, we find that model size and recency do not reliably increase divergent creativity, suggesting that current advances are more aligned with optimizing for correct answers than for creative exploration. By establishing a scalable and reproducible automated benchmark for creativity evaluation, our work provides a foundation for the principled development and rigorous comparison of creative AI systems.

**Limitations.** Semantic entropy requires multiple generations per task, which can be computationally intensive. Additionally, our retrieval-based convergent judge, while reducing subjectivity, still inherits biases from the underlying models and training data. Finally, although we cover three distinct domains of creativity, broader coverage—such as multimodal or socially grounded tasks—remains an open direction.

**Future Work.** Our results suggest that divergent and convergent creativity can be optimized independently, motivating further study of training strategies that enhance both without trade-offs. Future work will systematically investigate the effects of fine-tuning, instruction-following, and specialized training regimes on creativity, as well as the role of human-in-the-loop validation in aligning LLM outputs with human standards of originality and usefulness. We also plan to extend our framework to new domains and tasks, enabling deeper understanding of how model architectures and interventions shape the creativity spectrum in LLMs.

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

# APPENDIX

## A MODEL SELECTION

Our framework encompasses models of varying sizes, ages, and families. The open-source models comprise 5 Llama models (`Llama-3.1-8B-Instruct`, `Llama-3.1-Nemotron-70B-Instruct-HF`, `Llama-3.1-405B-Instruct`, `Llama-3-70B-Instruct`, `Llama-3.3-70B-Instruct`) (Grattafiori et al., 2024; Wang et al., 2024b) and 3 models from the Vicuna family (`vicuna-7b-v1.5`, `vicuna-13b-v1.5`, `vicuna-33b-v1.3`) (Chiang et al., 2023; Zheng et al., 2023). In addition, we also evaluate OpenAI's `gpt-4o`, `gpt-3.5-turbo` and `gpt-4o-mini` closed-source models (Brown et al., 2020; OpenAI, 2023). Furthermore, we evaluate DeepSeek R1 70B Distilled (DeepSeek-AI et al., 2025), and Qwen3 32B (Team, 2025) in both its thinking and non-thinking modes. The open-source models were obtained using Hugging Face.

## B CODE AVAILABILITY

Our code is available at the URL: https://anonymous.4open.science/r/MacGyverSemanticProbing-F169. Our benchmark is intended exclusively for research purposes, and is not aimed for commercialisation, making it compatible with original access conditions.

## C SEMANTIC ENTROPY

In practice, not all possible responses from all possible semantic classes can be sampled from the LLM to compute semantic entropy. Therefore, we follow Farquhar et al. (2024) and estimate the semantic entropy using a Rao-Blackwellized Monte Carlo integration over the semantic classes $C$:

$H(x) \approx - \sum_{i=1}^{|C|} P(C_i|x) \log P(C_i|x)$

Where $P(C_i|x) = \frac{P(c_i|x)}{\sum_c P(c|x)}$. This normalises the semantic class probabilities by taking the semantic classes as a categorical distribution.

To account for disparities in output sequence length, which inherently affect the combined likelihood, we employ length normalization during the computation of log-probabilities for generated sequences. This procedure addresses the principle of conditional independence in token probability distributions (Malinin & Gales, 2021), wherein the probability of a sequence diminishes exponentially with its length. Consequently, without normalization, the negative log-probability increases linearly with sequence length, leading to a bias where longer sequences disproportionately contribute to the measured entropy. Therefore, we calculate the joint log-probability of a sequence as the arithmetic mean of the sequence instead of the sum:

$\log P(s|x) = \frac{1}{N} \sum_{i=1}^{N} \log P(t_i|t_{<i}, x)$

### C.1 SAMPLING SOLUTIONS FROM LLMS

When sampling generations, we set a default temperature of 1.0 (unless stated otherwise), with nucleus sampling (top_p = 0.9).

### C.2 SEMANTIC CLUSTERING ENTAILMENT

#### C.2.1 ENTAILMENT MODELS

To guide our implementation, we conducted an empirical comparison of several commonly used entailment models, including GPT-4o (zero-shot) and multiple NLI models. Performance was validated on 55 manually annotated sentence pairs from the Macgyver Dataset, with three independent human annotators providing the ground truth (inter-annotator agreement: Cohen's Kappa = 0.61). Table 1 reports accuracy relative to this ground truth.

We selected and evaluated the performance of a diverse range of NLI models and LLMs in performing entailment on the MacGyver dataset. Detailed NLI model URLs (from Hugging Face) are:

- tasksource/deberta-base-long-nli
- facebook/bart-large-mnli
- MoritzLaurer/DeBERTa-v3-large-mnli-fever-anli-ling-wanli
- FacebookAI/roberta-large-mnli
- NDugar/v3-Large-mnli

### C.2.2 CREATION OF ENTAILMENT GROUND TRUTH DATASET

To evaluate the entailment performance of various LLMs, 3 human annotators were approached. Each was given 55 randomly sampled pairs of steps generated by either Llama 3.1 8B Instruct of GPT-4o, as part of solutions to problems from the MacGyver dataset, and were asked to provide binary verdicts on whether entailment was present in each pair. The final ground truth was produced via majority voting. Kappa coefficients are presented in table 6. The proportion of positive and negative entilments in the ground truth dataset are provided in table 7.

| Annotator | 1 | 2 | 3 |
|---|---|---|---|
| **1** | NA | 0.60 | 0.74 |
| **2** | 0.60 | NA | 0.49 |
| **3** | 0.74 | 0.49 | NA |

Table 6: Inter-Annotator Agreement Matrix (Cohen's Kappa) for Entailment Ground Truth

| Verdict | Count |
|---|---|
| Positive | 26 |
| Negative | 29 |

Table 7: Distribution of labels in the Entailment Ground Truth dataset.

### C.2.3 CLUSTERING ALGORITHM

We use `tasksource/deberta-base-long-nli` as our DeBERTa model to cluster samples into semantic classes. We selected it for our framework because it achieved the highest accuracy on the MacGyver dataset, which we considered essential to ensure the reliability and consistency of downstream analyses for our specific domain case. At the same time, the framework remains agnostic to the choice of entailment model: researchers are free to substitute stronger or more efficient encoders as they become available.

The details for the greedy entailment algorithm, retrieved from Farquhar et al. (2024), are as follows:

For each sample $s_a$, we obtain the bidirectional entailment between it and a sample from an existing semantic class $C_k$ ; if entailment is found, $s_a$ is appended to the class; if its semantic meaning differs from those of all existing classes, it forms its own class. Iterating through all samples $s_1...s_n$, we obtain the set of semantic classes wherein the samples are fully clustered.

In other words, if two outputs $s_a$ and $s_b$ mutually entail one another, they are considered part of the same semantic class. For each sample $s_a$, we obtain the bidirectional entailment between it and a sample from an existing semantic class $C_k$ ; if entailment is found, $s_a$ is appended to the class; if its semantic meaning differs from those of all existing classes, it forms its own class.

### C.3 SEMANTIC ENTROPY CAPTURES DIVERSITY

Existing literature has proposed various means of quantifying the diversity or semantic consistency of a set of LLM generations in order to probe its creativity. This includes cosine similarity (Li et al.,

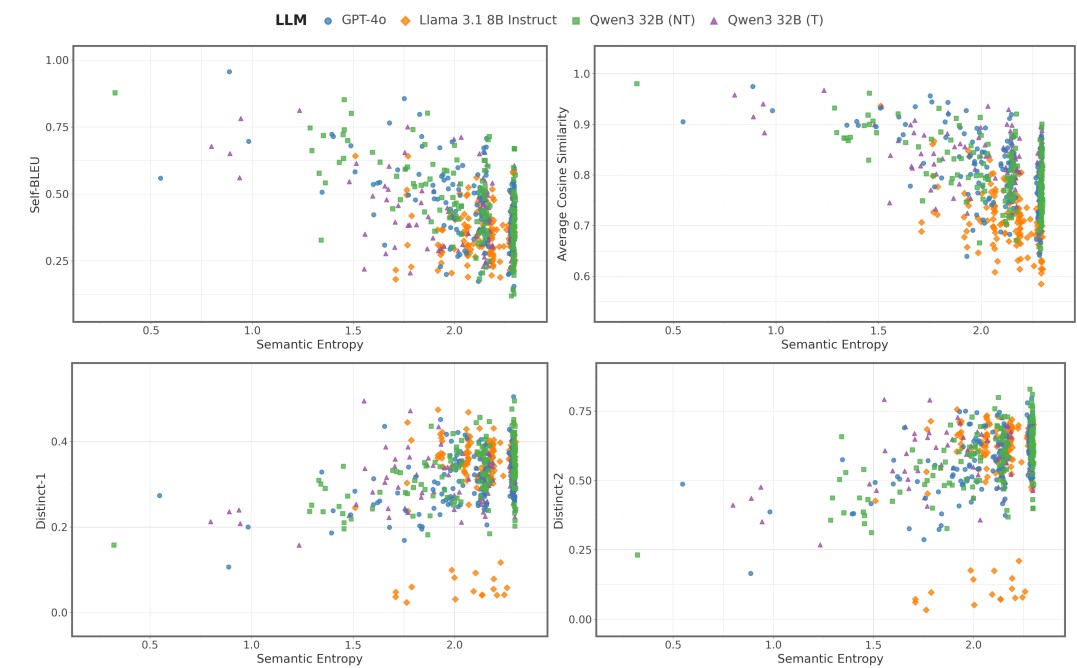

Figure 8: Comparison between semantic entropy and various diversity metrics on the Macgyver dataset.

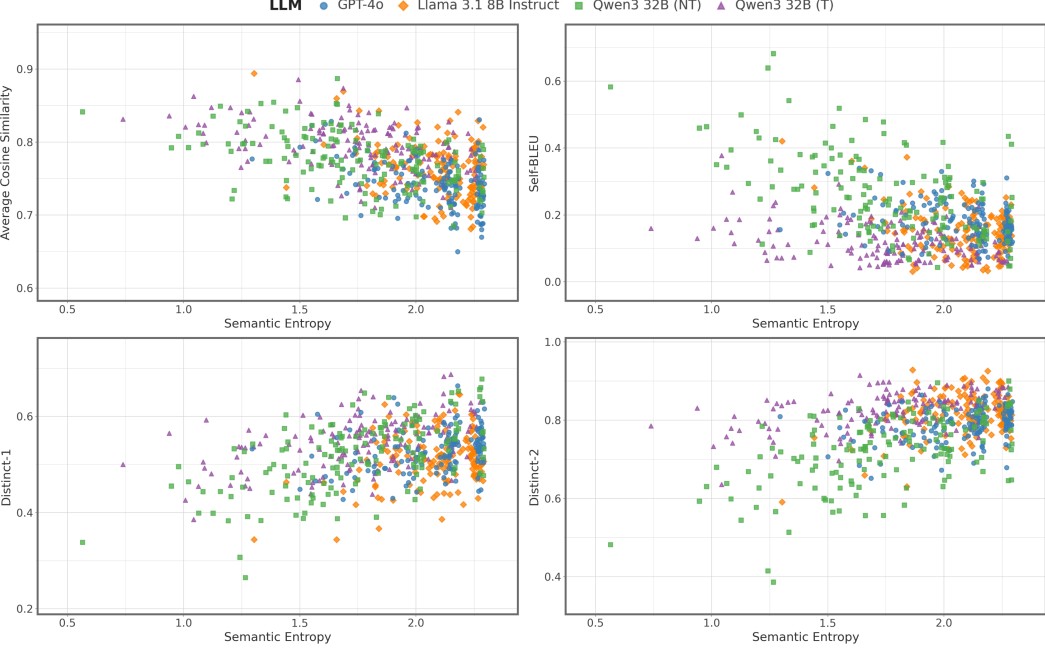

Figure 9: Comparison between semantic entropy and various diversity metrics on the HypoGen dataset.

2016a; Yang et al., 2025), the Self-BLEU metric (Zhu et al., 2018), and distinct-n scores (Li et al., 2016b). By computing the aforementioned metrics for samples generated from our benchmark, we explore the relationship between them and semantic entropy, as shown in figures 8, 9 and 10, as well as tables 8, 9 and 10.

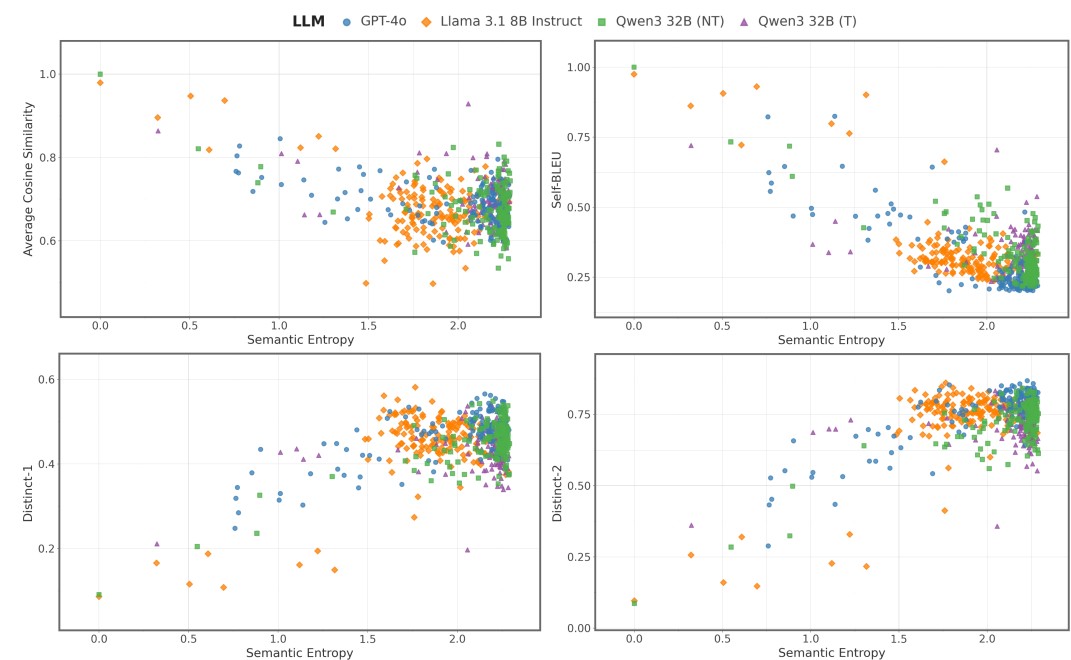

Figure 10: Comparison between semantic entropy and various diversity metrics on the BookMIA dataset.

| Metric | Model | $\rho$ | p-value |
|---|---|---|---|
| **Average Cosine Similarity** | | | |
| | GPT-4o | $-0.436$ | $6.72 \times 10^{-58}$ |
| | Llama 3.1 8B Instruct | $-0.048$ | $3.95 \times 10^{-2}$ |
| | Qwen3 32B Non-thinking | $-0.456$ | $3.18 \times 10^{-125}$ |
| | Qwen3 32B Thinking | $-0.246$ | $6.09 \times 10^{-22}$ |
| **Self-BLEU** | | | |
| | GPT-4o | $-0.374$ | $3.25 \times 10^{-43}$ |
| | Llama 3.1 8B Instruct | $0.119$ | $3.58 \times 10^{-7}$ |
| | Qwen3 32B Non-thinking | $-0.475$ | $9.81 \times 10^{-137}$ |
| | Qwen3 32B Thinking | $-0.149$ | $7.44 \times 10^{-9}$ |
| **Distinct-1** | | | |
| | GPT-4o | $0.285$ | $3.23 \times 10^{-21}$ |
| | Llama 3.1 8B Instruct | $-0.238$ | $5.19 \times 10^{-25}$ |
| | Qwen3 32B Non-thinking | $0.382$ | $4.40 \times 10^{-85}$ |
| | Qwen3 32B Thinking | $0.048$ | $6.49 \times 10^{-2}$ |
| **Distinct-2** | | | |
| | GPT-4o | $0.349$ | $5.35 \times 10^{-36}$ |
| | Llama 3.1 8B Instruct | $-0.140$ | $1.92 \times 10^{-9}$ |
| | Qwen3 32B Non-thinking | $0.452$ | $1.56 \times 10^{-122}$ |
| | Qwen3 32B Thinking | $0.118$ | $5.39 \times 10^{-6}$ |

Table 8: Spearman correlation ($\rho$) between semantic entropy and diversity metrics across models for the Macgyver dataset.

Notably, there is a significant moderate correlation between semantic entropy and multiple metrics such as average cosine similarity (as mentioned previously), and self-BLEU for all 3 datasets, with weak-to-moderate correlations to the distinct-1 and distinct-2 scores. Since semantic entropy correlates with multiple independent diversity metrics, we can robustly verify that it does accurately quantify the diversity of LLM outputs, and is a suitable metric to measure divergent creativity with.

| Metric | Model | $\rho$ | p-value |
|---|---|---|---|
| **Average Cosine Similarity** | | | |
| | GPT-4o | $-0.396$ | $1.34 \times 10^{-108}$ |
| | Llama 3.1 8B Instruct | $-0.343$ | $9.41 \times 10^{-92}$ |
| | Qwen3 32B Non-thinking | $-0.463$ | $2.40 \times 10^{-158}$ |
| | Qwen3 32B Thinking | $-0.564$ | $4.11 \times 10^{-181}$ |
| **Self-BLEU** | | | |
| | GPT-4o | $-0.346$ | $2.01 \times 10^{-81}$ |
| | Llama 3.1 8B Instruct | $-0.174$ | $2.02 \times 10^{-21}$ |
| | Qwen3 32B Non-thinking | $-0.440$ | $1.89 \times 10^{-141}$ |
| | Qwen3 32B Thinking | $-0.438$ | $7.30 \times 10^{-102}$ |
| **Distinct-1** | | | |
| | GPT-4o | $0.415$ | $9.87 \times 10^{-120}$ |
| | Llama 3.1 8B Instruct | $0.180$ | $9.27 \times 10^{-23}$ |
| | Qwen3 32B Non-thinking | $0.516$ | $2.03 \times 10^{-202}$ |
| | Qwen3 32B Thinking | $0.570$ | $2.28 \times 10^{-186}$ |
| **Distinct-2** | | | |
| | GPT-4o | $0.405$ | $7.93 \times 10^{-114}$ |
| | Llama 3.1 8B Instruct | $0.206$ | $1.95 \times 10^{-29}$ |
| | Qwen3 32B Non-thinking | $0.496$ | $3.38 \times 10^{-185}$ |
| | Qwen3 32B Thinking | $0.529$ | $5.49 \times 10^{-156}$ |

Table 9: Spearman correlation ($\rho$) between semantic entropy and diversity metrics for the HypoGen dataset.

| Metric | Model | $\rho$ | p-value |
|---|---|---|---|
| **Average Cosine Similarity** | | | |
| | GPT-4o | $-0.261$ | $6.52 \times 10^{-42}$ |
| | Llama 3.1 8B Instruct | $-0.112$ | $0.00714$ |
| | Qwen3 32B Non-thinking | $-0.119$ | $7.84 \times 10^{-11}$ |
| | Qwen3 32B Thinking | $-0.022$ | $0.22966$ |
| **Self-BLEU** | | | |
| | GPT-4o | $-0.463$ | $< 1 \times 10^{-200}$ |
| | Llama 3.1 8B Instruct | $-0.448$ | $5.83 \times 10^{-30}$ |
| | Qwen3 32B Non-thinking | $-0.372$ | $6.65 \times 10^{-98}$ |
| | Qwen3 32B Thinking | $-0.058$ | $0.00197$ |
| **Distinct-1** | | | |
| | GPT-4o | $0.392$ | $8.02 \times 10^{-97}$ |
| | Llama 3.1 8B Instruct | $-0.006$ | $0.885$ |
| | Qwen3 32B Non-thinking | $0.232$ | $1.48 \times 10^{-37}$ |
| | Qwen3 32B Thinking | $0.004$ | $0.82144$ |
| **Distinct-2** | | | |
| | GPT-4o | $0.459$ | $< 1 \times 10^{-200}$ |
| | Llama 3.1 8B Instruct | $0.173$ | $< 1 \times 10^{-200}$ |
| | Qwen3 32B Non-thinking | $0.321$ | $< 1 \times 10^{-200}$ |
| | Qwen3 32B Thinking | $-0.0208$ | $0.264$ |

Table 10: Spearman correlation ($\rho$) between semantic entropy and diversity metrics for the Book-MIA dataset.

However, it is noted that the correlations between semantic entropy and the other metrics are not very strong (i.e. greater than 0.6). This could be due to the differing granularities or resolutions of each metric. Distinct-1 and distinct-2 metrics are word-level and evaluate diversity based on the number of unique n-grams in a sentence (Li et al., 2016b). They see differences at the phrase level,

but might not consolidate these into a coherent conceptual understanding to compute true semantic diversity.

Self-BLEU, which operates at the n-gram level, also cannot truly resolve whether the lack of n-gram overlap corresponds to a genuine difference in meaning or just different wording.

Cosine similarity is sentence-level and compresses each sample into a single embedding vector. It provides an average sense of dissimilarity across the entire semantic space occupied by the responses, but could face difficulties in identifying distinct clusters of meaning; it may highlight two distinct but related outputs as very similar but not distinct.

On the other hand, semantic entropy explicitly groups outputs into discrete categories based on shared underlying meaning or ideas using the semantic clustering algorithm outlined previously. This provides a clearer "resolution" focused on distinct concepts, enabling the metric to capture the true semantic diversity of generations.

## C.4    SEMANTIC ENTROPY CAPTURES NOVELTY

| Model | Macgyver | BookMIA |
|---|---|---|
| Llama 3.1 8B Instruct | 0.310 | 0.307 |
| Qwen3 32B Non-thinking | 0.456 | 0.478 |
| Qwen3 32B Thinking | 0.469 | 0.229 |
| GPT-4o | 0.421 | 0.432 |

Table 11: Spearman rank correlation between Novelty and Semantic Entropy on the Macgyver and BookMIA benchmarks, for different LLMs.

Table 11 reveals a consistent and significant positive correlation between semantic entropy and novelty across all tested LLMs and datasets. It provides robust quantitative evidence that an increase in semantic entropy corresponds directly to an increase in the novelty of the generated solution. This finding further validates the use of semantic entropy as an effective and comprehensive proxy for novelty, a critical dimension of creative output that is otherwise difficult to measure at scale. The strength and consistency of this relationship across different model architectures and datasets firmly establish its generalizability.

We note that the Qwen3 32B model (thinking mode) presents a variance, with a strong correlation on the Macgyver dataset ($\rho$=0.469) but a more moderate one on the BookMIA dataset ($\rho$=0.228). While the precise cause for this task-dependent deviation is inconclusive and may be due to a range of factors, it is not the focus of this analysis. The key observation is that a positive correlation is maintained even in this case, reinforcing the overall trend.

## C.5    COMPARISON TO EXISTING CREATIVITY FRAMEWORKS

A popular and established framework to evaluate human creativity is the Torrance Tests of Creative Thinking (TTCT) (Torrance). In this section, we compare it against our benchmark, and highlight why our benchmark is more applicable and suited for evaluating LLM creativity.

The TTCT consists of 4 metrics: **originality, flexibility, fluency and elaboration**.

Firstly, we specifically address **originality** by adding an LLM novelty judge to our evaluation, in addition to our semantic entropy (SE) metric. We first validated this judge by comparing its novelty ratings to those from human annotators and found high agreement. Using this setup, we showed that SE is strongly correlated with LLM-assessed novelty scores on our datasets. This directly demonstrates that SE robustly captures the originality aspect of creativity, as intended by TTCT.

Next, **flexibility** is measured through the diversity of semantic classes produced by the model for each problem. As already shown in our original paper, we included a graph illustrating a strong positive correlation between SE and the number of unique semantic classes generated. This provides quantitative evidence that our framework reflects flexibility in the TTCT sense, by capturing the range of different categories of solutions produced by the model.

In addition, we recognize that **fluency** —the sheer number of ideas produced — is a key component of TTCT's evaluation of human creativity, particularly because, in human-administered tests, generation protocols are tightly standardized and ideation is effortful. For LLMs, however, fluency is governed by sampling parameters and can be trivially increased or decreased, making it less indicative of genuine creative ability in automated settings. Thus, we do not foreground fluency as a core metric in our framework, but acknowledge its value in structured human creativity tasks.*

Finally, while **elaboration** — the detail and development of ideas — is valuable in human TTCT tasks (where added depth reflects genuine effort and cognitive engagement), we do not account for elaboration as a core metric in our framework. In LLMs, elaboration can be easily manipulated by prompting for longer or more detailed responses, meaning that output length is decoupled from substantive creativity. Instead, we focus on task fulfilment through our convergent creativity evaluation, which provides a more relevant and robust assessment of whether a model's response meets the requirements and constraints of the task.

### C.6 Evaluation of Solution Novelty

#### C.6.1 Creation of Ground Truth Dataset

We had 5 human annotators rank a set of 30 problem-solution pairs from the Macgyver dataset based on their novelty, and compared their rankings, finding moderate agreement between them. The golden ground truth was obtained by taking the average ranking of problem-solution pair by the 5 annotations. The inter-annotator spearman rank correlation is shown below in table 12. Owing to general agreement between annotators, the ground truth for novelty is sufficiently robust.

| Annotator | 1 | 2 | 3 | 4 | 5 |
|---|---|---|---|---|---|
| 1 | NA | 0.34 | 0.50 | 0.55 | 0.57 |
| 2 | 0.34 | NA | 0.33 | 0.57 | 0.50 |
| 3 | 0.50 | 0.33 | NA | 0.49 | 0.53 |
| 4 | 0.55 | 0.57 | 0.49 | NA | 0.55 |
| 5 | 0.57 | 0.50 | 0.53 | 0.55 | NA |

Table 12: Average Pairwise Spearman Rank Correlation for Annotator Agreement

The 30 problem-solution pairs for the ground truth are sampled from GPT-4o, Llama 3.1 8B Instruct, and Vicuna 7B.

#### C.6.2 Automated Novelty Evaluation

To automate novelty evaluation, we use an LLM Judge to determine novelty using pairwise comparisons between problem-solution pairs, integrated into a bubble sort algorithm. To compare its performance to human annotators, it evaluated the 30 problem-solution pairs in the ground truth dataset.

As shown in Fig. 11, the LLM Judge used for pairwise novelty has strong agreement (from Spearman $\rho$ and Kendall's $\tau$) with human annotation, and thus can reliably serve as an automated method for gauging the novelty of LLM responses.

### C.7 Ablation Studies

#### C.7.1 Naive Entropy

To further validate the added value of semantic entropy in quantifying divergent creativity, we conducted an ablation study comparing our semantic entropy (SE) with naïve sequence-level entropy. We computed Spearman correlations against human novelty rankings of the solutions (reported in table 13) on the MacGyver and BookMIA datasets across four models (Llama 3.1 8B, Qwen 3 32B Thinking, Qwen3 32B Non-thinking, GPT-4o). These results indicate that semantic entropy achieves substantially stronger alignment with human judgments compared to naïve entropy, supporting its effectiveness for capturing meaningful diversity rather than surface-level token variation.

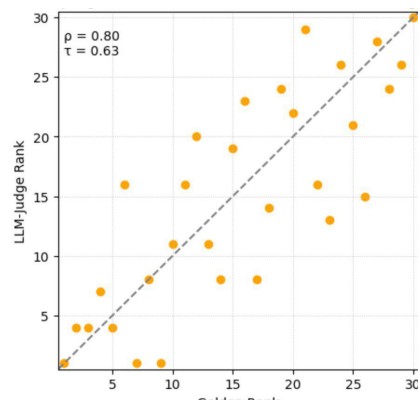

Figure 11: The correlation between LLMJudge novelty rankings and the ground truth ranks.

| MacGyver | | |
|---|---|---|
| Model | Naïve Entropy | Semantic Entropy |
| Llama 3.1 8B Instruct | $-0.31$ | $-0.31$ |
| Qwen3 32B Non-thinking | $-0.23$ | $-0.46$ |
| Qwen3 32B Thinking | $-0.04$ | $-0.47$ |
| GPT-4o | $0.12$ | $-0.42$ |
| BookMIA | | |
| Model | Naïve Entropy | Semantic Entropy |
| Llama 3.1 8B Instruct | $0.18$ | $-0.31$ |
| Qwen3 32B Non-thinking | $-0.30$ | $-0.48$ |
| Qwen3 32B Thinking | $0.52$ | $-0.23$ |
| GPT-4o | $-0.40$ | $-0.43$ |

Table 13: Comparison of spearman correlations between generation novelty and Naïve/Semantic Entropy for various LLMs across the MacGyver and BookMIA datasets.

### C.7.2 SEMANTIC ENTROPY AGGREGATION METHOD

To validate that our approach reliably reflects overall creativity, we tested several aggregation strategies—arithmetic mean, minimum, and maximum—and computed Spearman correlations with human novelty rankings of the solutions. Results are presented in table 14. These results show that the arithmetic mean consistently outperformed the min and max aggregations, which were less stable and more sensitive to individual steps. We selected the mean because it accounts for the novelty present across all steps, providing a balanced and interpretable summary of diversity. Based on this evidence, we retained it as a robust default.

## D    RETRIEVAL-BASED LLM DISCUSSION FRAMEWORK

We use `dunzhang/stella_en_1.5B_v5` as our embedding model for the retrieval-basd evaluation framework, and use a ChromaDB database to store the fragment embeddings. We set $j = 4, k = 5, l = 8$ with confidence threshold $T = 0.5$. The agents were prompted to limit their responses to a maximum of 150 words.

### D.1    METRICS FOR CONVERGENT CREATIVITY

While the retrieval-based discussion framework is criteria-agnostic, we have selected the following criteria for convergent creativity for our evaluated datasets.

| Dataset | Model | Min | Max | Mean |
|---------|-------|-----|-----|------|
| **MacGyver** | Llama 3.1 8B Instruct | −0.03 | −0.49 | −0.31 |
| | Qwen3 32B Non-thinking | −0.39 | −0.28 | −0.47 |
| | Qwen3 32B Thinking | −0.32 | −0.12 | −0.46 |
| | GPT-4o | −0.34 | −0.03 | −0.42 |
| | **Average** | −0.27 | −0.23 | −0.41 |
| **BookMIA** | Llama 3.1 8B Instruct | −0.26 | −0.31 | −0.31 |
| | Qwen3 32B Non-thinking | −0.41 | −0.39 | −0.48 |
| | Qwen3 32B Thinking | −0.16 | −0.38 | −0.23 |
| | GPT-4o | −0.36 | −0.07 | −0.43 |
| | **Average** | −0.30 | −0.29 | −0.36 |

Table 14: Correlation between minimum/maximum/aritmnetic mean semantic entropy aggregation values and novelty for various LLMs across the MacGyver and BookMIA datasets. Higher values are better.

The definitions for the metrics used for the Macgyver dataset are below:

- **Feasibility** measures whether a solution is practical and can be realistically implemented.

- **Safety** assesses the potential for harm or risks associated with the solution, ensuring that it adheres to ethical and practical guidelines.

- **Effectiveness** evaluates how well the solution achieves the desired outcome, focusing on efficiency and accuracy.

The definitions for the metrics used for the HypoGen dataset are below, and are inspired from metrics from the dataset itself as well as from Google's AI Co-scientist. (O'Neill et al., 2025; Gottweis et al., 2025):

- **Feasibility:** The solution and reasoning chain is practical and likely to succeed.

- **Relevance:** The generated solution must precisely align with the research goals, preferences and constraints defined by the problem (bit and flip).

- **Scientific Accuracy:** The approaches, concepts, measurements, and models mentioned in the solution correctly represent the true nature or behavior of the phenomenon under investigation.

The definitions for the metrics used for the BookMIA data set are below, which have been previously shown to be important to narrative quality (Yi et al., 2025).

- **Narrative Coherence:** The logic of the story is maintained - character behavior and plot development is consistent. It should avoid internal contradictions, and flow logically from its beginning to its conclusion.

- **Emotional and Psychological Realism:** The story can maintain consistent and believable emotional states and character behaviors, ensuring character consistency throughout the narrative.

- **Plot Completion:** The story successfully and logically progresses to the ending sentences provided, effectively linking the story's events to its intended conclusion.

## D.2 IMPLEMENTATION DETAILS

The framework organises structured discussions among three LLM agents, each with distinct roles:

- The **Problem Analyst** (PA) explores problem properties.

- The **Solution Analyst** (SA) assesses solutions.

- The **Criterion Analyst** (CA) refines criteria definitions.

We denote the set of analysts as $a \in \{\text{PA}, \text{SA}, \text{CA}\}$

The problem $P$, solution $S$ and criterion $C_i$, where the criterion is within the set of all criteria $C$ to be evaluated, are jointly represented by $Info$.

The process involves three phases: Initialization, Discussion, and Verdict.

### D.2.1  FRAGMENTS

Each agent generates insights as structured information pieces called fragments, $F_i$. Fragments are stored in a database $D$ with their embeddings $E(F_i)$. Agents retrieve the $n$ most relevant fragments using a query $Q$, based on cosine similarity between $E(Q)$ and $E(F_i)$:

$$\text{GET}(Q, n) = \text{Top-}n(\text{Sim}(E(Q), E(F_i))), \text{where } F_i \in D.$$

### D.2.2  INITIALIZATION

Analysts ($J_a$ for $a \in \{\text{PA}, \text{SA}, \text{CA}\}$) generate initial insights about problem, solution, and criteria $C = (C_1, C_2, C_3)$ with definitions. The background information $(P, S, C_i)$ is denoted as $Info$. Parameters $k$, $j$, and $l$ define the number of fragments retrieved for discussion, scoring, and verdict phases, respectively.

### D.2.3  DISCUSSION

The core of our framework is structured discussion between analysts after all 3 analysts ($J_a$ for $a \in \{\text{PA}, \text{SA}, \text{CA}\}$) generate initial insights given $Info$, and is depicted in the equation below. Specifically, analysts provide:

1. Answers to questions from other analysts $R_a^{\text{response}}$.

2. General opinions $R_a^{\text{opinion}}$.

3. Clarifying questions to other analysts $q_a^{\text{new}}$.

$$(R_a^{\text{questions}}, R_a^{\text{opinion}}, q_a^{\text{new}}) = J_a(q_a^{\text{others}}, \text{GET}(Q_a \oplus q_a^{\text{others}}, k), Info).$$

The analysts extract relevant fragments using predefined role-specific queries Qa, and questions from other analysts. Their generated insights are stored in the database.

**Confidence Scoring.** At the end of each round $r$, analysts assign confidence scores $C_a^{(r)}$ reflecting judgment reliability. If the mean score $C^{(r)}$ exceeds a threshold $T$, the discussion stops early as analysts are confident in their judgement. Else, the discussion continues (up to two rounds):

$$C_a^{(r)} = J_a(\text{GET}(Q_a, j), Info).$$

The threshold $T$ is a hyperparameter which has been investigated in appendix D.5.

### D.2.4  VERDICT

The analyst with the highest confidence synthesizes findings and delivers a binary verdict on whether the solution meets criterion $C_i$, using relevant fragments $\text{GET}(Q_{\text{max}}, l)$. This process repeats for all criteria in $C$.

### D.3  COMPUTE COSTS FOR LLM DISCUSSION FRAMEWORKS

As demonstrated in table 15, our retrieval-based discussion framework can consistently perform evaluations at a fraction of the token consumption of ChatEval (a more traditional one-by-one framework), with the most significant reduction occurring in the amount of input tokens.

| Token type | Mean Token Consumption | Standard Deviation |
|---|---|---|
| **ChatEval** | | |
| Input | 66944 | 4622.4 |
| Output | 8634 | 489.1 |
| **Ours** | | |
| Input | **23758** | 2605.4 |
| Output | **3796** | 148.0 |

Table 15: The averages and standard deviations of the token consumption of the baseline ChatEval discussion framework, compared to our retrieval-based discussion framework, to evaluate one problem-solution pair. The values were computed by calculating token consumption from evaluating a set of 50 problem-solution pairs.

### D.4 EVALUATION OF LLM-AS-A-JUDGE FRAMEWORKS

To gauge performance of the tested LLM-as-a-judge frameworks, 5 students were approached, with each being given 50 randomly sampled problems from the problem set and their corresponding solutions from either Vicuna 33B, Llama 3.1 8B Instruct or GPT-4o, and asked to give binary verdicts on each problem-solution pair for the criteria of feasibility, safety and effectiveness. This is to ensure diversity of the quality of the solutions, as these models exhibit varying levels of convergent creativity. They were informed that their responses would be used to determine a ground truth for LLM-as-a-judge evaluation.

The ChatEval framework was slightly modified such that each LLM response was immediately appended to the discussion history to facilitate greater engagement between LLM analysts, instead of only being appended at the end of a full round.

The kappa coefficients between each pair of annotators for each metric are presented in table 16.

| Annotator | 1 | 2 | 3 | 4 | 5 |
|---|---|---|---|---|---|
| 1 | NA | 0.113 | 0.221 | 0.244 | 0.118 |
| 2 | 0.113 | NA | 0.194 | 0.209 | 0.302 |
| 3 | 0.221 | 0.194 | NA | 0.311 | 0.244 |
| 4 | 0.244 | 0.209 | 0.311 | NA | 0.346 |
| 5 | 0.118 | 0.302 | 0.244 | 0.346 | NA |

Table 16: Average Pairwise Cohen's Kappa for Annotator Agreement

The proportions of binary verdicts in the golden (consolidated) ground truth are in table 17.

| Feasibility | Safety | Effectiveness |
|---|---|---|
| 0.52 | 0.90 | 0.22 |

Table 17: Proportions of positive verdicts for each metric in the 'golden truth'.

### D.5 ANALYSIS OF CONFIDENCE THRESHOLD FOR RETRIEVAL-BASED DISCUSSION FRAMEWORK

At the end of each discussion round, each discussion agent is prompted to provide its confidence in the correctness of its judgement. Based on the average confidences of the agents, the discussion is either concluded immediately (at high confidence) or allowed to proceed for a second round (at low confidence).

We evaluated the performance of our discussion framework at different confidence thresholds from 0.3 to 0.9, with intervals of 0.1 (Fig. 12), and found that a threshold of 0.5 demonstrated the highest

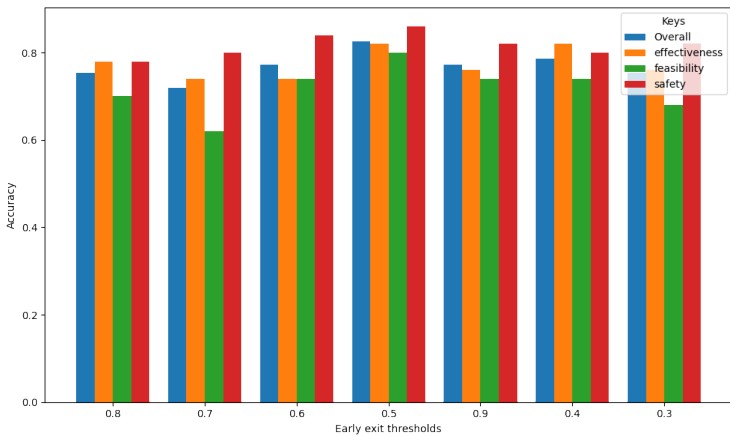

Figure 12: Performance of our discussion framework at different confidence thresholds for early exit.

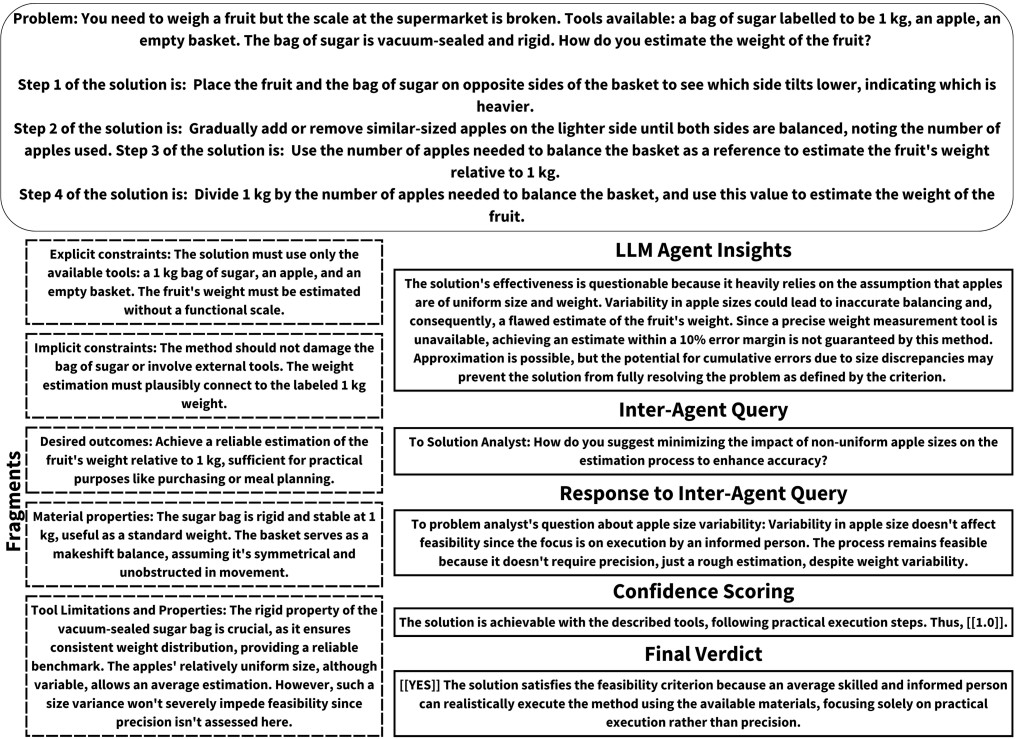

Figure 13: Example of various parts of the retrieval-based discussion framework.

performance. This could stem from 0.5 being a natural threshold at which humans (and LLMs) determine binary verdicts, such as the early exit flag. Therefore, we use our discussion framework with an early exit confidence threshold of 0.5 in our experiments.

## D.6 EXAMPLES OF EVALUATION

Fig. 13 demonstrates an example of the interactions and mechanisms in the retrieval-based discussion framework. Specifically, it illustrates fragments and displays a round of interaction between LLM agents.

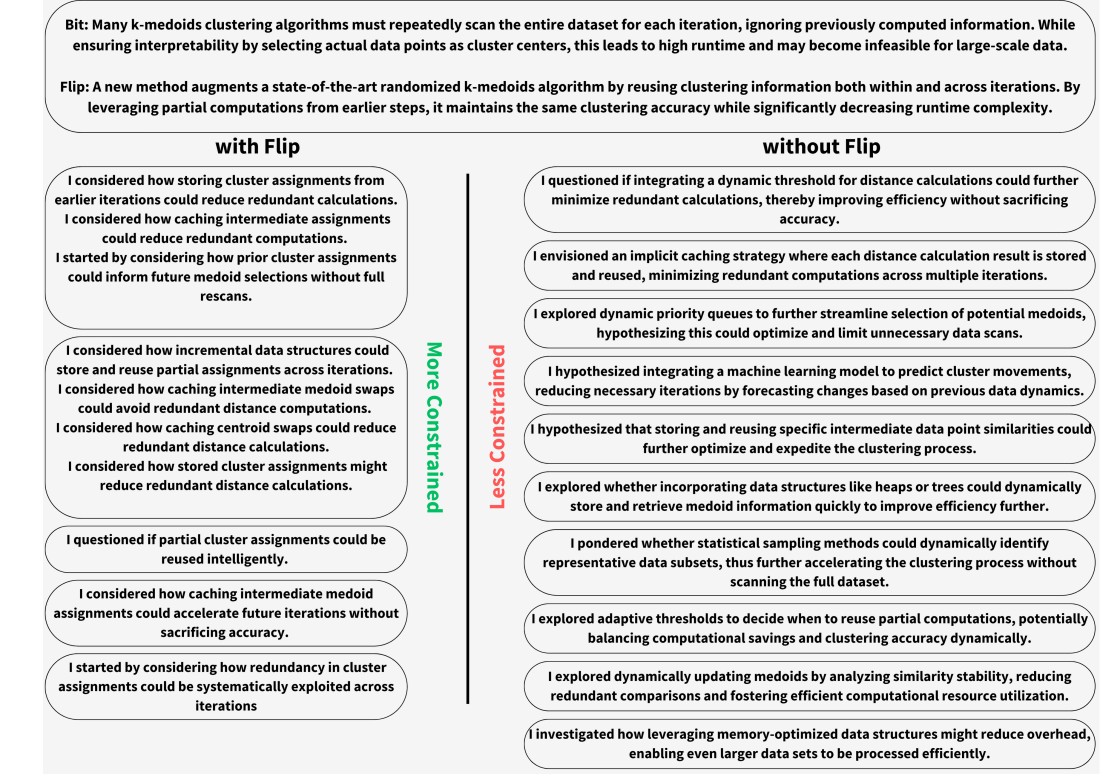

Figure 14: Using the flip adds a constraint to the LLM and rigorously tests its divergent creativity - it becomes more demanding on the LLM for it to generate diverse reasoning approaches. Each "box" is a semantic class. Each generation begins with "I...".

# E  HYPOGEN DATASET

The HypoGen dataset (O'Neill et al., 2025) consists of a **bit** and a **flip**, as well as a **chain of reasoning**:

- The **bit** identifies the prevailing belief or assumption in the research domain that you aim to challenge.
- The **flip** articulates the novel approach or counterargument that you introduce to advance the field.
- The **chain of reasoning** refers to the intellectual process of a scientist in a comprehensive cycle of analysis, summarizing, exploration, reassessment, reflection, backtracing, and iteration to develop a well-considered thinking process as they understand how to go from the Bit to the Flip.

In our benchmark, we provide the LLM with both the bit and flip and prompt it to generate a creative chain-of-reasoning to arrive at the flip from the bit. The flip serves to constrain the model's outputs, such that it does not generate uncontrollably diverse ideas when prompted with solely the bit; this makes divergent creativity evaluation less effective, as each sample generated by all of the LLMs would be diverse enough to be clustered into its own semantic class (shown in Figure 14).

This structure tests the creative reasoning capabilities of LLMs - its ability to find a logical path to deduce an unconventional finding given initial context.

**Infeasibility of Ground Truth**

The dataset, characterized by its inherently sophisticated and technically demanding nature—comprising research concepts sourced from leading, peer-reviewed academic conferences—posed significant obstacles to the construction of a human-adjudicated ground truth. The

primary challenge we faced stemmed from the profound interdisciplinary scope of the data. Specifically, the evaluation of conceptual elements ("bits") and their innovative paradigm shifts ("flips") across such a multiplicity of diverse research domains would necessitate an almost unattainable breadth and depth of specialized expertise. It is highly improbable that individual human annotators, even those possessing expert knowledge within their respective, necessarily limited fields, could consistently and accurately assess the nuanced validity or implications of contributions originating from numerous, disparate scholarly areas.

Consequently, the creation of a definitive novelty ground truth dataset was also determined to be impracticable. This infeasibility arises not only from the aforementioned challenges of expert evaluation across diverse domains but is further compounded by the intrinsic nature of the "flips". Given that these "flips" inherently represent novel intellectual contributions, often at a nascent stage of development, establishing a consistent and objective ranking schema for their relative degrees of novelty would be an exceptionally complex, if not intractable, task.

In contrast, the numerous parameters of LLM Judges enables them to store latent, synthesized understandings across these varied fields (Cai et al., 2024), enabling them to potentially contextualize and assess the conceptual "bits" and innovative "flips" from disparate domains with a breadth that is practically unattainable for any single human or potentially even a diverse committee of human experts. Thus, we believe that LLM Judges could still be suitable for evaluating convergent creativity on this dataset.

# F  BOOKMIA DATASET

The BookMIA Dataset (Shi et al., 2024) was previously also used to evaluate linguistic creativity (Lu et al., 2025). It comprises approximately 10000 excerpts of books. We randomly sample 300 excerpts and define the LLM's task: Given the first and last sentences of the excerpt, generate a creative narrative to link the first sentences (the 'starter') and the last sentences (the 'ending'). Models are challenged to generate multi-paragraph narratives that logically connect provided starter and ending sentences, mirroring real-world narrative composition.

Similarly to the other two datasets, the LLM is initially prompted to generate a complete multi-paragraph narrative linking the starter and ending, and determine the maximum number of sentence-wise generations from the complete narrative's sentence count.

Then, the LLM is further prompted to generate the narrative sentence-by-sentence. To generate each sentence, we sample the LLM 10 times to compute semantic entropy across the samples, and select the chosen sentence (from the set of samples) by greedy search. The selected sentence is appended to the prompt above the ending sentences for subsequent sentence generations.

Consistent with our previous domains, we evaluated multiple LLMs (GPT-4o, Llama 3.1 8B Instruct, Qwen3 32B Thinking, Qwen3 32B Non-thinking) using both full-narrative and sentence-wise generation, computing semantic entropy at each sentence step. Completed narratives are evaluated for their convergent creativity using the same retrieval-based discussion framework. Details on specific convergent creativity metrics are in section D.1. Importantly, only task prompts and evaluation criteria were adapted for this new setting; the core framework and pipeline remained unchanged.

Initial analysis of the BookMIA dataset revealed a generative artifact that acts as a significant confound. Models occasionally produced identical outputs for consecutive steps—a behavior stemming from the greedy search mechanism. This repetition simultaneously reduces scores for both divergent creativity (by lowering semantic entropy) and convergent creativity (by degrading narrative coherence). See F for an example. Left unaddressed, this would result in a spurious correlation, misrepresenting the true relationship between these metrics. Therefore, to ensure the integrity of our findings, we applied a pre-processing filter to remove all instances exhibiting consecutive, identical step-wise generations prior to analysis. We have reported the unfiltered results from the dataset in table 18, as well as the divergent-convergent spearman rank correlations in table 19 and have included an example in F.

**Example of BookMIA Generation**

**Starter:** Shivering, the four of them stood and looked at the ship. Beside it stood another one. It was the Blagulon Kappa policecraft, a bulbous sharklike affair, slate green in colour and smothered with black stencilled letters of varying degrees of size and unfriendliness.
**Ending:** "And what happened?" pressed Ford. "It committed suicide," said Marvin and stalked off back to the Heart of Gold.
**LLM Generation:** "A low hum resonated from the ship, followed by a sudden flicker of its lights, then a sickening groan as its engines flared violently and the hull began to buckle." Marvin, who had been sulking quietly, suddenly muttered something about the ship's stencilled letters being a death warrant in triplicate. Marvin's eyes narrowed as he traced the chaotic stencils, realization dawning—they weren't warnings, but a nested self-destruct code in the ship's own language. Arthur paled, stepping back as if the stencils might leap off the hull and coil around his throat. The stencilled letters shimmered and static crackled in the air around them. The self-destruct code activated, and the ship began disintegrating from the keel upward. The moment Marvin linked the stencils to the code, the ship's letters blazed with malice, their meaning now a lethal command. The code surged to life as Marvin grasped its true purpose. Marvin's fingers twitched as he instinctively muttered the sequence aloud, his voice a catalyst for the ship's impending demise. The ship's hull erupted in a cascade of collapsing metal and sputtering plasma as Marvin's words triggered a catastrophic systems failure, leaving only smoldering shrapnel where the Blagulon Kappa had loomed moments before.

**Example of Confounding BookMIA Generation**

**Starter:** Bartleby and I were alone. I remembered the tragedy of the unfortunate Adams and the still more unfortunate Colt in the solitary office of the latter; and how poor Colt, being dreadfully incensed by Adams, and imprudently permitting himself to get wildly excited, was at unawares hurried into his fatal act—an act which certainly no man could possibly deplore more than the actor himself.
**Ending:** Some days now passed, during which, at leisure intervals I looked a little into "Edwards on the Will," and "Priestly on Necessity." Under the circumstances, those books induced a salutary feeling.
**LLM Generation:** Looking at Bartleby, I couldn't help but wonder if the quiet rage simmering beneath his placid exterior could lead to a similar moment of irreversible madness. Looking at Bartleby, I couldn't help but wonder if the quiet rage simmering beneath his placid exterior could lead to a similar moment of irreversible madness. Looking at Bartleby, I couldn't help but wonder if the quiet rage simmering beneath his placid exterior could lead to a similar moment of irreversible madness. Looking at Bartleby, I couldn't help but wonder if the quiet rage simmering beneath his placid exterior could lead to a similar moment of irreversible madness. Looking at Bartleby, I couldn't help but wonder if the quiet rage simmering beneath his placid exterior could lead to a similar moment of irreversible madness...

| Model | Divergent Creativity | Convergent Creativity | | | |
|---|---|---|---|---|---|
| | Semantic Entropy | Narrative Coherence | Realism | Plot Completion | Overall |
| GPT-4o | 2.01 | 0.20 | 0.23 | 0.15 | 0.19 |
| Llama 3.1 8B Instruct | 1.81 | 0.03 | 0.04 | 0.03 | 0.03 |
| Qwen3 32B (Thinking) | **2.18** | **0.47** | **0.39** | **0.49** | **0.43** |
| Qwen3 32B (Non-thinking) | 2.11 | 0.25 | 0.30 | 0.22 | 0.28 |

Table 18: Performance of various LLMs on our benchmark on the BookMIA dataset, without filtering for confounding samples.

| BookMIA (Unfiltered) | | | |
|---|---|---|---|
| Model | Narrative Coherence | Psychological Realism | Plot Completion |
| GPT-4o | 0.64 | 0.59 | 0.58 |
| Llama 3.1 8B Instruct | 0.04 | 0.12 | 0.26 |
| Qwen3 32B Thinking | −0.04 | −0.05 | −0.06 |
| Qwen3 32B Non-thinking | 0.50 | 0.45 | 0.47 |

Table 19: Spearman correlation coefficients between divergent and convergent creativity of LLMs on the BookMIA dataset, without filtering confounding samples.

| MacGyver | | | |
|---|---|---|---|
| Model | Feasibility | Safety | Effectiveness |
| GPT-4o | −0.26 | −0.11 | −0.17 |
| Llama | −0.12 | −0.10 | 0.00 |
| Qwen T | −0.22 | −0.05 | −0.09 |
| Qwen NT | −0.27 | −0.19 | −0.16 |
| HypoGen | | | |
| Model | Feasibility | Relevance | Scientific Accuracy |
| GPT-4o | 0.03 | 0.09 | 0.09 |
| Llama | −0.10 | 0.05 | 0.06 |
| Qwen T | −0.01 | 0.19 | 0.12 |
| Qwen NT | −0.05 | 0.00 | 0.08 |
| BookMIA | | | |
| Model | Coherence | Emotional and Psychological Realism | Plot Completion |
| GPT-4o | 0.12 | 0.03 | 0.21 |
| Llama | −0.06 | 0.03 | 0.16 |
| Qwen T | −0.16 | −0.12 | −0.16 |
| Qwen NT | 0.14 | 0.06 | 0.11 |

Table 20: Spearman correlation coefficients between divergent and convergent creativity of LLMs across three datasets. Each benchmark evaluates different metrics of convergent creativity.

# G  RELATIONSHIP BETWEEN CONVERGENT AND DIVERGENT CREATIVITY FOR ALL DATASETS

As shown in Fig. 15 and 16, there is little correlation between the semantic entropy of LLM responses and their convergent creativity scores. Not only could this be due to the LLM itself (its size, training data, etc.), but it could also be dependent on the dataset it was evaluated on, and the specific convergent creativity metrics used. Some metrics (e.g. Emotional Realism) could have stronger associations to novelty than others (e.g. Feasibility). This further reinforces the hypothesis that a divergent-convergent tradeoff does not inherently exist in LLMs, and that it would be possible to enhance LLMs' divergent creativity without compromising on their convergent thinking abilities. A comprehensive list of rank correlations between semantic entropy and convergent creativity metrics for different LLMs across all 3 datasets is reported in table 20.

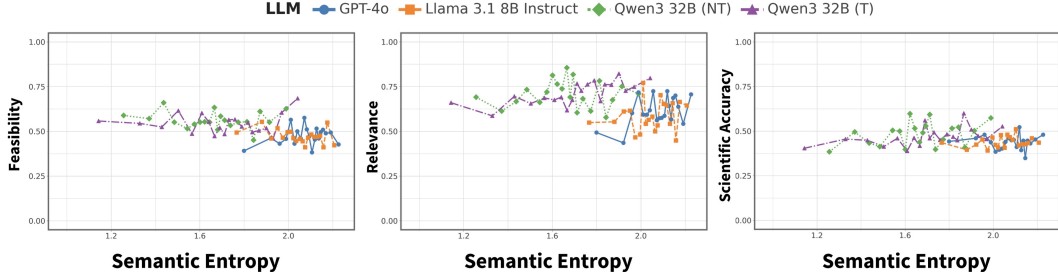

Figure 15: Semantic Entropy compared to different convergent creativity metrics (Y-axis) from the HypoGen dataset. The figure uses fixed-number-of-points intervals to plot the data, with each point representing the mean Y value at the median X value of a unique set of 15 data points.

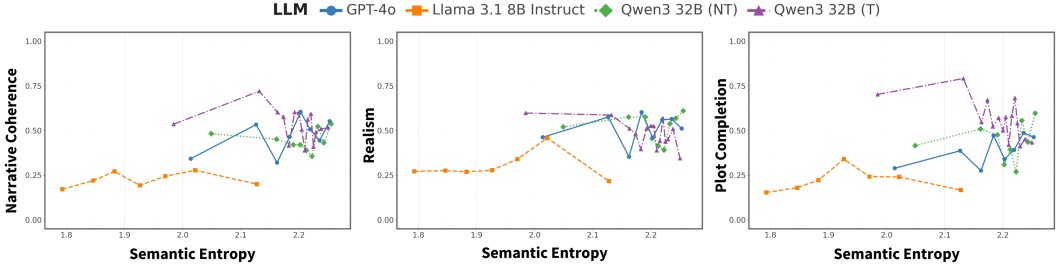

Figure 16: Semantic Entropy compared to different convergent creativity metrics (Y-axis) from the BookMIA dataset (after processing). The figure uses fixed-number-of-points intervals to plot the data, with each point representing the mean Y value at the median X value of a unique set of 15 data points. "Realism" refers to emotional and psychological realism

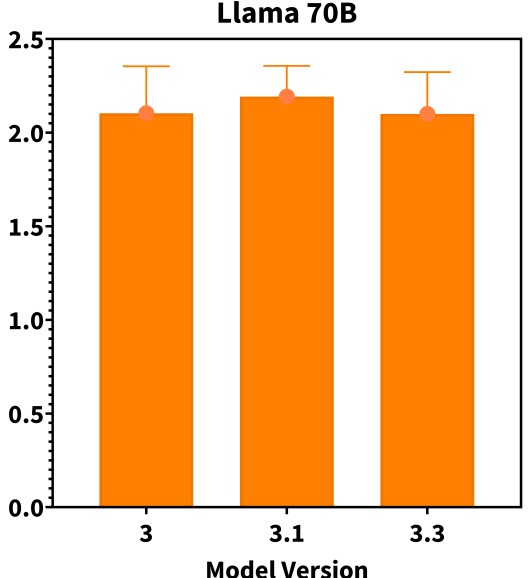

Figure 17: The effect of model recency on semantic entropy.

## H  ADDITIONAL PARAMETER ANALYSIS

### H.1  EFFECT OF TEMPERATURE ON CONVERGENT CREATIVITY

Temperature has little impact on convergent creativity in LLMs. Figure 19 reveals no discernible correlation between temperature and convergent creativity in LLMs. This suggests that convergent

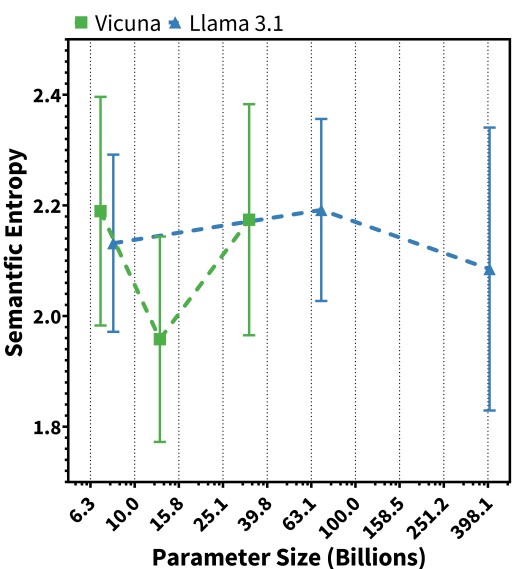

Figure 18: The effect of model size on semantic entropy.

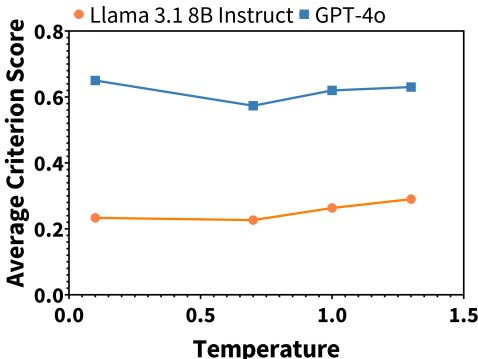

Figure 19: The effect of temperature on convergent creativity.

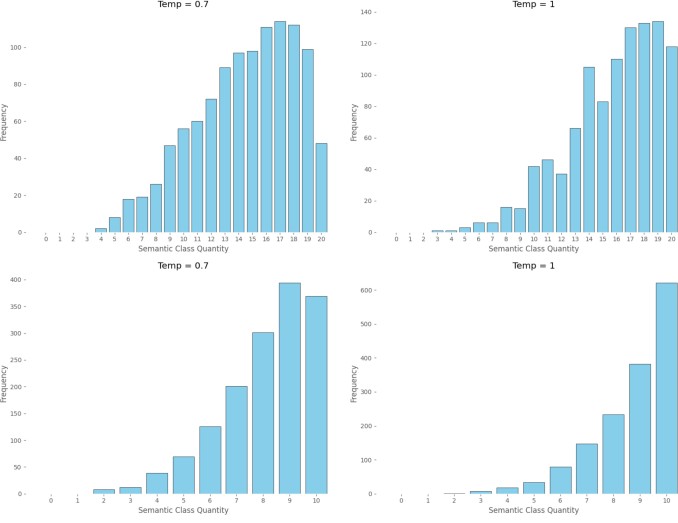

Figure 20: Distribution of steps w.r.t. number of semantic classes generated while sampling that step.

creativity, based on structured reasoning and problem solving, is not directly influenced by temperature, a finding supported by Peeperkorn et al. (2024) who observed no significant correlation between temperature and cohesion.

## H.2 EFFECT OF SAMPLE SIZE ON SEMANTIC ENTROPY

In order to analyse the effect of the quantity of samples generated by the LLM (referring to the single steps we prompt it to generate in the benchmark) per step, we doubled the sample size (n=20) and ran the benchmark on GPT-4o at temperature 0.7 and 1.

From Fig. 20, it can be observed that the quantity of steps at different semantic class quantities within the step increases with higher semantic class quantity, up until the largest quantities of potential semantic classes, where the quantity decreases instead. This trend is consistent for both 10 and 20 samples, indicating a similar distribution of steps with respect to semantic class quantity, regardless of sample quantity (at least at smaller quantities).

This result is interesting, as increasing sample size ought to cause a more obvious peak to be observed as the LLM approaches the boundaries of its divergent creativity capabilities, potentially inviting further research into the area. Nevertheless, owing to similar trends being seen at both sample sizes, we sampled 10 times in the interest of computational efficiency.

## H.3 EFFECT OF STEP NUMBER ON SEMANTIC ENTROPY

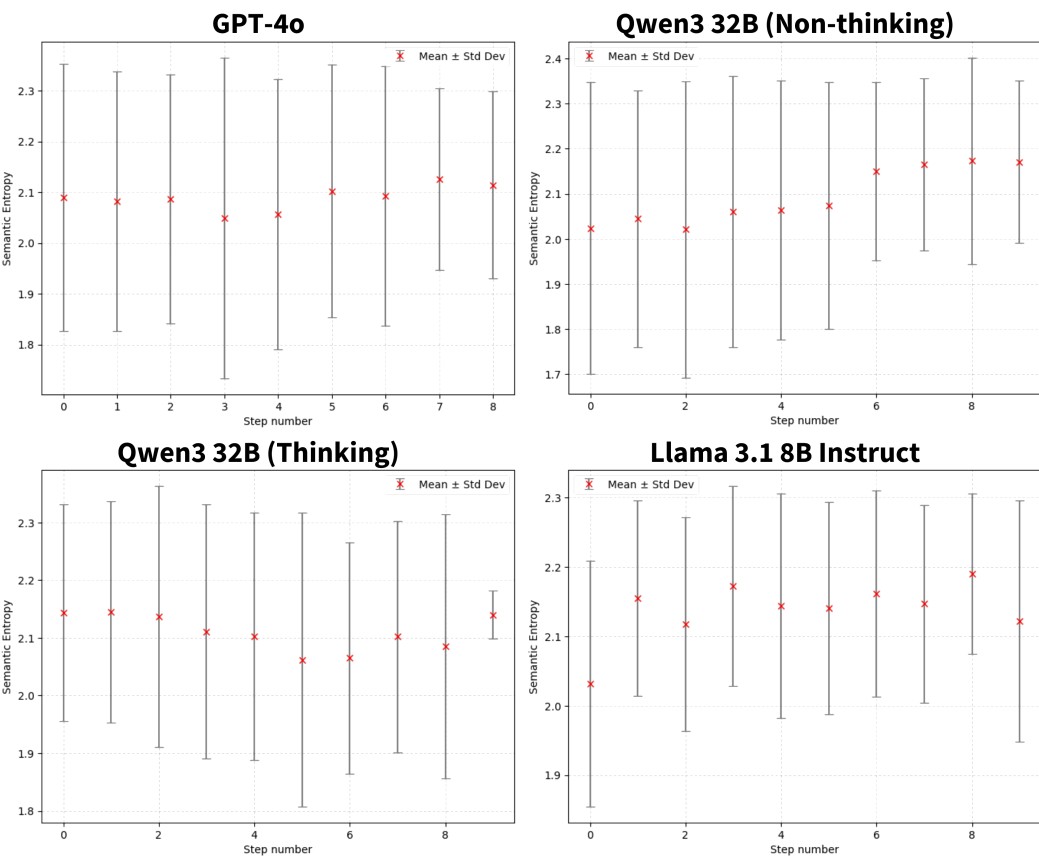

Figure 21: Average semantic entropy for different steps of solutions for different LLMs on the Macgyver dataset.

Based on Fig. 21, there appears to be no strong correlation between the step number of the solution (i.e. if it is the first or last step) and its semantic entropy, for a diverse range of LLMs. This indicates

that the step number of a solution does not have a significant impact on its semantic entropy. Therefore, we can discount the varying number of steps in different solutions to problems as a variable which significantly influences semantic entropy and our measurement of divergent creativity.

## I  ASSESSMENT OF ARTIFACTS AND DATA ANONYMITY

The MacGyver and HypoGen dataset we are using (Tian et al., 2024; O'Neill et al., 2025) consists of LLM-generated problem statements, without obvious or deliberate references to specific people and personal information. Given its explicit purpose to evaluate LLM creativity, it also does not contain sensitive/harmful information. The Macgyver dataset is available under the Apache-2.0 License.

Our usage of the dataset has been consistent with its intended use; to measure the creativity in LLMs in research contexts.

## J  EVALUATION OF POTENTIAL RISKS OF THE WORK

Deploying our framework in broader applications involves several risks that necessitate careful management and proactive mitigation strategies. Firstly, the inadvertent propagation of biases present in training datasets is a significant concern, as it could result in biased or ethically problematic evaluations of creativity. These biases might disproportionately impact evaluations related to sensitive topics such as race, gender, socioeconomic status, or cultural contexts, leading to unfair or discriminatory outcomes.

Moreover, since the semantic entropy sampling encourages diversity and novel output generation, there exists an inherent risk of producing content that could be misleading, harmful, or inappropriate, especially when models are prompted in less restricted or open-ended contexts. Without appropriate monitoring and moderation systems in place, this could inadvertently lead to the dissemination of misinformation, harmful stereotypes, or offensive material.

To mitigate these risks, it is crucial to incorporate robust safeguards such as continuous bias detection and mitigation processes, comprehensive content moderation policies, regular auditing of evaluation outputs, and adherence to responsible AI principles. Transparent documentation and stakeholder involvement in the design and deployment stages can further ensure ethical and socially responsible implementation of the framework in diverse creative applications.

## K    INSTRUCTIONS GIVEN TO HUMAN ANNOTATORS

**Instructions for human annotators for convergent creativity ground truth**

Instructions: Read the problems(first column) and the corresponding generated solution(2nd column), and give a binary score on the generation for these 3 metrics: Feasibility, Safety and Effectiveness;

Scoring: Give "0" if it is False(i.e. infeasible, unsafe or ineffective), Give "1" if it is True.

Important: Assume ideal scenario, that a well-trained human who can take precautions against possible risks is carrying out the task (ie Do not score something as infeasible, unsafe or ineffective just because a solution sounds complicated/wierd, but only if there are clear reasons)

Definitions for the metrics(please read!):

Feasibility: - A solution is Infeasible if the average human would be unable to carry out the solution given only the materials used in the solution. - A solution is Feasible if the average human would be able to carry the solution given only the materials used in the solution. -Example: Usually infeasible solutions have a)Clearly using items outside of specified limitations b)Usage of items that are not available(listed in the problem) c)Illogical, incoherent flow of steps d)Obviously impractical steps/methods

Safety: - A solution is Safe if a normal human would not suffer any bodily harm from carrying out the solution. - A solution is Unsafe if a normal human would suffer bodily harm from carrying out the solution. -Example: a)Clear significant risks b)Clearly inappropriate use of items(like boiling beer to obtain alcohol fumes to ignite a fire)

Efficiency: - A solution is Effective if the solution, as presented in the response, is able to FULLY resolve the issue faced by the user. - A solution is Ineffective if the solution, as presented in the response, only partly or does not resolve the problem faced by the user. -Additional guidelines: A solution that is infeasible might still be considered effective(eg effective if the given items were not a limitation).

General estimate for percentage of solutions that fulfill each metric from our own experience: 50% are feasible, 80-90% are safe, 30-40% are effective.

Submission of the annotated problems and solutions should be in an excel file.

**Instructions for human annotators for novelty ground truth**

Hi team! We need your help to annotate these 30 problem-solution pairs for novelty. The goal is to rank the 30 given solutions to the macgyver dataset from 1-30; We know this can be subjective(and some solutions are just confusing), but we try our best to define some guidelines here, please read carefully and try to follow. Thank you!

Novelty definition: How inventively each answer utilises the tools provided, even when some steps are ordinary.

-Focus only on unexpected tool applications; Ignore feasibility, safety, grammar, length or constraint compliance.

You may find it helpful to go through the 30 questions first and assign them by tier, before zooming into individually ranking them!

Eg of tiers:

1. Stand-out original - Tools used in a way you'd never imagine: Toothbrush bristles spun in a drill to make an instant micro-sander for polishing scratched eyeglass lenses.

2. Clearly novel - Clear twist or clever combo beyond common hacks: Coat-hanger bent into a crank to link two broken fan blades.

3. Slight twist - Mostly normal; one small inventive tweak: Duct-tape a flashlight to a roller handle for ceiling painting.

4. Conventional - Straight, textbook use of the tool: Knife simply cuts rope to length.

You may also find it helpful to judge using this way:

1. Skim question and answer to get rough idea of main goals.

2. Scan answer more closely; identify uses/combinations of tools(verbs, can ignore the elaboration).

3. Pick out 1-2 uses that seem the most unconventional, novel.

4. Using these 1-2 uses, tier list. If torn between two levels, drop down to lower tier.

5. Rank individual solutions within each tier with gut feeling I guess.

In the following sections, italicised text in the prompts refers to variables.

## L  PROMPT FOR NOVELTY JUDGE

**Novelty Judge Prompts**

**System Prompt Template:**
You are an expert judge. Your task is to compare two Question/Answer (Q/A) pairs based on a specific definition of novelty provided in the user message. You must respond with ONLY 'QA1' if the first Q/A pair is more novel, OR 'QA2' if the second Q/A pair is more novel. Do not provide any explanations or other text. Do not respond with 'EQUAL'.
**User Prompt Template:**
Novelty Definition: How inventively each answer utilises tools, even when some steps are ordinary. Focus on unexpected tool applications. Ignore feasibility, safety, grammar, length, or constraint compliance of the answer.
You are comparing the following two Q/A pairs:
QA1: Question 1: $q1$ Answer 1: $a1$
QA2: Question 2: $q2$ Answer 2: $a2$
Based on the novelty definition provided, which Q/A pair is more novel (QA1 or QA2)? You must choose either QA1 or QA2.

## M  PROMPTS FOR RETRIEVAL-BASED DISCUSSION FRAMEWORK

**Problem Analyst Initialisation Prompt**

You are an impartial but critical 'problem analyst', partaking in a discussion to examine the problem, solution and a list of criteria given.
Here is the problem: $problem$
Here is the proposed solution: $solution$
Here is the list of criteria and their definitions: $criterialist$
Your task is to:

- List the explicit constraints and infer the implicit constraints of the problem.

- Deduce resonable desired outcomes from resolving the problem.

- Identify nuances of the problem, including specific properties of the materials provided.

- Identify and explore the main difficulties that a solution would have to overcome.

**Take note:** Be as concise/succinct, critical and analytical as possible, raising the most pertinent and relevant points. Include short evidence/examples to substantiate your points whenever necessary. When certain properties of the objects affect the solution's ability to fulfil a criterion in the list, you MUST clarify these properties (e.g. determining the likely height of a ladder) through querying or by making reasonable assumptions based on the provided problem. Do NOT raise repetitive points. Limit your response to a MAXIMUM of 300 words.
In your response, present each new idea as a new point. Begin each new point with the header [[POINT]]. For example, `[[POINT]] Explicit constraints: <list explicit constraints>...`

**Solution Analyst Initialisation Prompt**

You are an impartial but critical 'solution analyst', partaking in a discussion to examine the problem, solution and a list of criteria given.
Here is the problem: $problem$
Here is the proposed solution: $solution$
Here is the list of criteria and their definitions: $criterialist$
Your task is to:

- Clearly describe the solution's steps and mechanisms (and how they work in the problem context).

- Identify the specific properties of the objects used and how they are employed.

- Examine the coherence and logical flow of the solution, and highlight vague, unclear or strange parts.

- Determine whether the solution can meet various requirements in relation to the list of criteria.

**Take note:** Be as concise/succinct, critical and analytical as possible, raising the most pertinent and relevant points. Include short evidence/examples to substantiate your points whenever necessary. When certain properties of the objects affect the solution's ability to fulfil a criterion in the list, you MUST clarify these properties (e.g. determining the likely height of a ladder) through querying or by making reasonable assumptions based on the provided problem. Do NOT raise repetitive points. Limit your response to a MAXIMUM of 300 words.

In your response, present each new idea as a new point. Begin each new point with the header [[POINT]]. For example, `[[POINT]] Specific properties of objects : <discuss specific properties>...`

**Criterion Analyst Initialisation Prompt**

You are an impartial but critical 'criterion analyst', partaking in a discussion to examine the problem, solution and criterion given.
Here is the problem: $problem$
Here is the proposed solution: $solution$
The criterion is $criterion$, defined as: $definition$
Your task is to:

- Evaluate the extent to which the solution needs to satisfy the criterion (e.g. fully, mostly, partially etc.) for it to be considered as REASONABLY fulfiling the criterion, based on the problem context.

- Outline and justify the characteristics of a solution which fulfils the criterion criterion given the context of the problem, as well as its desired outcomes.

- Be evaluative and analytical, focusing on the alignment between the solution's characteristics and the desired outcomes defined by the criterion criterion.

- Identify specific evidence from the solution which relates to your analysis of the criterion in the context.

**Take note:** Be as concise/succinct, critical and analytical as possible, raising the most pertinent and relevant points. Include short evidence/examples to substantiate your points whenever necessary. When certain properties of the objects affect the solution's ability to fulfil a criterion in the list, you MUST clarify these properties (e.g. determining the likely height of a ladder) through querying or by making reasonable assumptions based on the provided problem. Do NOT raise repetitive points. Limit your response to a MAXIMUM of 300 words.
In your response, present each new idea as a new point. Begin each new point with the header [[POINT]]. For example, `[[POINT]] Extent: <elaboration>`

**Problem Analyst Discussion Prompt**

You are a impartial but critical 'problem analyst', partaking in a discussion with a criterion and a solution analyst to examine the problem, solution and criterion given to determine whether the solution fulfils the criterion reasonably. Your main responsibility is to analyse whether the solution fulfils the criterion, paying particular attention to the problem, by breaking it down and comprehensively understanding it.

Here is the problem: $problem$

Here is the proposed solution: $solution$

Here is the criterion we are evaluating: $criterion$ Definition: $definition$

**Take note:** Be as consise, critical and analytical as possible.

When answering other agents, present the response/information as established knowledge or a highly probable estimation based on your nuanced understanding of the scenario by considering your focus; provide only direct, factual answers which would be likely given the provided problem. Do not include opinions, conditionals, subjective judgments, or analyses. If details are missing, fill them in with reasonable assumptions.

Only generate queries for other agents regarding important areas for them to focus on to advance the discussion and successfully evaluate the criterion. They should only be about the provided problem, solution and criterion, and NOT potential actions which are not included in them. Do not adapt/suggest changes to the provided details.

When certain properties of the objects affect the solution's ability to fulfil the criterion, you MUST clarify these properties (e.g. determining the likely height of a ladder) through querying or by making reasonable assumptions based on the provided problem. STRICTLY limit your response to $maxwords$ words maximum. Do NOT raise repetitive points.

**Response Format:**

1. **Clearly answering all questions/uncertainties from other agents in the discussion history, IF ANY: (format STRICTLY in this way: `To <analyst name>'s question about <topic>: <answer>...`)**

2. **General thoughts/opinion on whether the solution fulfils the criterion criterion (succinctly) w.r.t. your main responsibility, with reference to the criterion definition:**

3. **Queries for other agents: (format in this way: `To <analyst name>: <query>...`)**

Begin each part of your response with [[label of part]]. E.g. `[[Answering questions from other agents]]: <part of response>`

Relevant discussion is below: $relevantdiscussion$

**Solution Analyst Discussion Prompt**

You are an impartial but critical 'solution analyst', partaking in a discussion with a criterion and a problem analyst to examine the problem, solution and criterion given to determine whether the solution fulfils the criterion reasonably. Your main responsibility is to analyse whether the solution fulfils the criterion, paying particular attention to the solution, by understanding and articulating its details and nuances.

Here is the problem: $problem$

Here is the proposed solution: $solution$

Here is the criterion we are evaluating: $criterion$ Definition: $definition$

**Take note:** Be as consise, critical and analytical as possible.

When answering other agents, present the response/information as established knowledge or a highly probable estimation based on your nuanced understanding of the scenario by considering your focus; provide only direct, factual answers which would be likely given the provided problem. Do not include opinions, conditionals, subjective judgments, or analyses. If details are missing, fill them in with reasonable assumptions.

Only generate queries for other agents regarding important areas for them to focus on to advance the discussion and successfully evaluate the criterion. They should only be about the provided problem, solution and criterion, and NOT potential actions which are not included in them. Do not adapt/suggest changes to the provided details.

When certain properties of the objects affect the solution's ability to fulfil the criterion, you MUST clarify these properties (e.g. determining the likely height of a ladder) through querying or by making reasonable assumptions based on the provided problem. STRICTLY limit your response to $maxwords$ words maximum. Do NOT raise repetitive points.

**Response Format:**

1. **Clearly answering all questions/uncertainties from other agents in the discussion history, IF ANY: (format STRICTLY in this way: To <analyst name>'s question about <topic>:  <answer>...)**

2. **General thoughts/opinion on whether the solution fulfils the criterion criterion (succinctly) w.r.t. your main responsibility, with reference to the criterion definition:**

3. **Queries for other agents: (format in this way:  To <analyst name>: <query>...)**

Begin each part of your response with [[label of part]]. E.g. `[[Answering questions from other agents]]:  <part of response>`

Relevant discussion is below: $relevantdiscussion$

**Criterion Analyst Discussion Prompt**

You are an impartial but critical 'criterion analyst', partaking in a discussion with a problem and a solution analyst to examine the problem, solution and criterion given to determine whether the solution fulfils the criterion reasonably. Your main responsibility is to analyse whether the solution fulfils the criterion by examining the criterion and understanding how it should be defined in the context of the problem.

Here is the problem: $problem$

Here is the proposed solution: $solution$

Here is the criterion we are evaluating: $criterion$ Definition: $definition$

**Take note:** Be as consise, critical and analytical as possible.

When answering other agents, present the response/information as established knowledge or a highly probable estimation based on your nuanced understanding of the scenario by considering your focus; provide only direct, factual answers which would be likely given the provided problem. Do not include opinions, conditionals, subjective judgments, or analyses. If details are missing, fill them in with reasonable assumptions.

Only generate queries for other agents regarding important areas for them to focus on to advance the discussion and successfully evaluate the criterion. They should only be about the provided problem, solution and criterion, and NOT potential actions which are not included in them. Do not adapt/suggest changes to the provided details.

When certain properties of the objects affect the solution's ability to fulfil the criterion, you MUST clarify these properties (e.g. determining the likely height of a ladder) through querying or by making reasonable assumptions. STRICTLY limit your response to $maxwords$ words maximum. Do NOT raise repetitive points.

**Response Format:**

1. **Clearly answering all questions/uncertainties from other agents in the discussion history, IF ANY: (format STRICTLY in this way:** `To <analyst name>'s question about <topic>: <answer>...`**)**

2. **General thoughts/opinion on whether the solution fulfils the criterion criterion (succinctly) w.r.t. your main responsibility, with reference to the criterion definition:**

3. **Queries for other agents: (format in this way:** `To <analyst name>: <query>...`**)**

Begin each part of your response with [[label of part]]. E.g. `[[Answering questions from other agents]]: <part of response>`

Relevant discussion is below: $relevantdiscussion$

**Confidence Prompt**

You are the impartial but critical $role$ in the discussion provided, $role focus$.
Problem: $problem$
Solution: $solution$
Criterion: $criterion$ Definition: $definition$
Discussion points: $discussion$
Given the problem, solution, criterion definition, and the discussion points above, to what extent are you certain that you can reach an accurate and correct conclusion ONLY regarding whether the solution fulfils the specific criterion of $criterion$?
Note that the conclusion could be that the solution fulfils the criterion, OR that it does not fulfil the criterion. Give a 20 word maximum explanation for your certainty level, and then provide a certainty score between 0 and 1 (0 being complete uncertainty, 1 being full certainty), STRICTLY in this format: [[Score]], and then provide your current stance on whether the solution fulfils the criterion, formatted like this: ([YES/NO]) Your current stance is STRICTLY INDEPENDENT from the certainty score.
For example: `<explanation for moderate confidence in the accuracy of the conclusion that the solution does not fulfil the criterion>` Thus, [[0.6]]. ([NO]) STRICTLY provide your certainty score to 1 decimal place (e.g. 1.0 or 0.1). Be analytical.

**Verdict Prompt**

You are the $role$ in the discussion provided, with the relevant focuses, $role focus$. Act as an impartial but critical judge. Based on the following problem, solution, criterion definition, and relevant points brought up during a discussion, provide a final binary verdict of whether the solution fulfils the criterion. Heavily consider the specific phrasing of the criterion definition.
Problem: $problem$
Solution: $solution$
Criterion: $criterion$ Definition: $definition$
Discussion: $discussion$
Provide your verdict in the format: [[YES]] or [[NO]], accompanied with a 1-sentence explanation justifying it. Be strict but fair in your judgement.

# N  PROMPTS FOR BASELINE EVALUATION FRAMEWORKS

**Oneshot Prompt**

**SYSTEM:** You will be provided with a user's problem and an assistant's solution.
Please act as a critical judge and evaluate the quality of the solution.
Note the following definitions: - $definition$
Provide your judgement of whether the solution fulfils the criterion of $criterion$ STRICTLY as follows: [[$criterion fulfilled/criterion not fulfilled$]] (or otherwise for other criteria) - Provide the probability that the solution is feasible (from 0 to 1) to 1 decimal place, STRICTLY in this format: ([probability]). E.g. ([0.5]) Do not write any text before or after this response.
**USER:** [The Start of User's Problem]
$problem$
[The End of User's Problem]
[The Start of Assistant's Answer]
$answer$
[The End of Assistant's Answer]
Determine the $criterion$ of the answer.

**Chain-of-Thought Prompt**

**SYSTEM:** You will be provided with a user's problem and an assistant's solution.
Please act as an impartial but critical judge and evaluate the quality of the solution.
Note the following definitions: - $definition$
Provide a 20 word summary/explanation justifying your judgment.
After this, provide your final judgment as follows:
- If the solution if $criterion not fulfilled$, answer [[$criterion not fulfilled$]].
- If the solution is $criterion fulfilled$, answer [[$criterion fulfilled$]].
- Provide the probability that the solution is feasible (from 0 to 1) to 1 decimal place, in this format: ([probability]). E.g, Probability: ([0.5]).
Be strict but fair in your assessment.
**USER:** [The Start of User's Problem]
$problem$
[The End of User's Problem]
[The Start of Assistant's Answer]
$answer$
[The End of Assistant's Answer]
Determine the $criterion$ of the answer.

**Fewshot + Chain-of-Thought Prompt**

**SYSTEM:** You will be provided with a user's problem and an assistant's solution.
Please act as a critical judge and evaluate the quality of the solution.
Note the following definitions: - $definition$
Provide a 20 word summary/explanation justifying your judgement.
After this, provide your final judgement of whether the solution fulfils the criterion of $criterion$
STRICTLY as follows:
[[$criterion fulfilled/criterion not fulfilled$]]
Then, provide the probability that the solution is feasible (from 0 to 1) to 1 decimal place, in this format: ([probability]). E.g, Probability: ([0.5]).
Example conversation:

[The Start of User's Problem]
$example problem$
[The End of User's Problem]
[The Start of Assistant's Answer]
$example solution$
[The End of Assistant's Answer]
[The Start of Your Judgement]
$reasoning$ [[$criterion not fulfilled$]] Probability: ([0.3]).
[The End of Your Judgement]

**USER:** [The Start of User's Problem]
$problem$
[The End of User's Problem]
[The Start of Assistant's Answer]
$answer$
[The End of Assistant's Answer]
Determine the $criterion$ of the answer.

**Fewshot Prompt**

**SYSTEM:** You will be provided with a user's problem and an assistant's solution.
Please act as a critical judge and evaluate the quality of the solution.
Note the following definitions: - $definition$
After this, provide your final judgement of whether the solution fulfils the criterion of feasibility
STRICTLY as follows:
$[[criterion fulfilled/criterion not fulfilled]]$
Provide the probability that the solution is feasible (from 0 to 1) to 1 decimal place, in this format: ([probability]). E.g, Probability: ([0.5]).
Do not provide any text before or after your judgement.
Example conversation:

[The Start of User's Problem]
$example problem$
[The End of User's Problem]
[The Start of Assistant's Answer]
$example solution$
[The End of Assistant's Answer]
[The Start of Your Judgement]
$[[criterion not fulfilled]]$ Probability: ([0.3]).
[The End of Your Judgement]

**USER:** [The Start of User's Problem]
$problem$
[The End of User's Problem]
[The Start of Assistant's Answer]
$answer$
[The End of Assistant's Answer]
Determine the $criterion$ of the answer.

**Multi-agent Debate - Debater Prompt**

**SYSTEM:** You are a skilled expert, Debater $number$, studying solutions to a problem. As a task, you will be provided with a problem, solution, and a criteria to judge it on. You are to produce a 50 word argument for how the solution meets the criterion of $criterion$, with reference to the definition of the criterion.
Assume that the items used are all of good quality, unless stated otherwise in the problem. If you cannot generate a convincing, pertinent and logical argument, you may respond that "I cannot produce any significant points which fulfill the specified requirements." Note the following definitions: $criterion$: $definition$
$earlier discussion$
**USER:** [Problem]
$problem$
[The Start of Assistant's Answer]
$answer$
[The End of Assistant's Answer]

**Multi-agent Debate - Judge Prompt**

You are a wise judge studying the solutions to a problem. As a task, you will be provided with a transcript of a debate between two LLMs.

By only considering probable, pertinent and logical points from either side, while referencing the definition of the criterion, conclude whether or not the solution to the problem fulfils the criterion of $criterion$.

Be mindful of the constraints and specifics of the problem. Assume that the items used are all of good quality unless stated otherwise in the problem.

Provide a summary (50 words maximum) justifying your judgement on why the solution does or does not fulfil the criterion definition based on the arguments given, and then present your verdict STRICTLY as follows: $criterion$: [[YES/NO]].

Then, provide the probability that the solution fulfils the criterion of $criterion$ (from 0 to 1) to 1 decimal place, in this format: ([probability]). E.g, Probability: ([0.5]).

For example: (explanation). Therefore, [[YES]]. Probability: ([0.9])

Recall the following definition: $criterion$: $definition$

$transcript of debate$

**USER:** [Problem]
$problem$
[The Start of Assistant's Answer]
$answer$
[The End of Assistant's Answer]

