# OpenReview forum: "Automated Creativity Evaluation for LLMs with Semantic Entropy and Efficient Multi-Agent Judging Across Open-Ended Tasks"
_ICLR.cc/2026/Conference — ICLR 2026 Conference Withdrawn Submission_

### Official Review · Reviewer_GDEb · 2025-10-27

**Soundness:** 3
**Presentation:** 3
**Contribution:** 3
**Rating:** 6
**Confidence:** 4

**Summary:**

The paper proposes a benchmark for evaluating model innovativeness and feasibility by simultaneously considering divergent and convergent creativity. Experimental results show that, compared to other baselines, the proposed method aligns more closely with human evaluations.

**Strengths:**

1. The paper evaluates models along two dimensions, measuring not only divergent creativity but also the feasibility of proposed solutions.
2. Experiments demonstrate that the proposed method correlates more strongly with human judgments than existing baselines.

**Weaknesses:**

See Questions below

**Questions:**

1. I noticed that Qwen3’s “thinking” score is not particularly high in the experiments. However, during the thinking process, models often explore many candidate solutions. Does the evaluation method treat the entire thinking trace as a single output, rather than evaluating each individual attempt as a separate path?
2. Model performance often varies across problems of different difficulty levels—for instance, on harder tasks, models typically require multiple rounds of trial and error before arriving at a correct solution. Would it be beneficial to evaluate models separately on problems of varying difficulty?
3. Different “thinking” models often exhibit substantial performance differences. Could the paper include results from a broader set of thinking models? Additionally, is it possible to correlate these models’ creativity scores with their capabilities in domains like math or code?
4. The paper states that divergent and convergent creativity stem from distinct mechanisms and can be optimized independently. However, does this independence depend on the problem type? For problems with very few viable solutions, achieving a high divergent creativity score may be inherently difficult—and attempting to artificially boost divergent creativity might compromise convergent performance.
5. Could the proposed method be adapted in the future as a reward signal to enhance model diversity during training or inference?

---

> ### Author Response · Authors · 2025-12-03
> **Response to Reviewer GDEb [1/3]**
>
> **Question 1: Evaluation of Reasoning Thinking Traces**
> >*"I noticed that Qwen3’s “thinking” score is not particularly high in the experiments. However, during the thinking process, models often explore many candidate solutions. Does the evaluation method treat the entire thinking trace as a single output, rather than evaluating each individual attempt as a separate path?"*
>
> ### **Response:**
>
> We sincerely thank the reviewer for their thoughtful and constructive feedback and appreciate your observation regarding the behavior of reasoning-enhanced models like Qwen3 and the nuances of evaluating creativity across varying difficulty levels.
>
> For models that generate internal reasoning traces (such as Qwen3-Thinking), we **explicitly remove the thinking tokens** and extract only the final, non-thinking output for analysis. We calculate Semantic Entropy by sampling a fixed quantity of generations, stripping the reasoning traces from each, and then clustering the *final answers* based on semantic equivalence.
>
> **Why Qwen3-Thinking shows lower Divergent Creativity:**
>
> Our empirical results (e.g., Table 3) show that Qwen3-Thinking achieves a lower Semantic Entropy score on the MacGyver dataset, compared to its Non-thinking counterpart. This is not necessarily a failure of the model, but rather a reflection of the nature of reasoning-enhanced fine-tuning:
>
> 1. The "thinking" process in these models is optimized to discard incorrect or implausible paths and converge on a robust solution. While the model may explore options internally within a single trace, the final distribution of outputs across samples tends to be more consistent (less entropic) because the reasoning process steers the model toward a narrow set of "optimal" solutions.
> 2. This aligns with our findings in Section 6.2, where we observe that while Qwen3-Thinking has slightly lower Divergent Creativity (semantic entropy), it scores significantly higher on Convergent Creativity. This confirms that the "thinking" process effectively acts as a filter, sacrificing raw diversity for higher task fulfillment.
>
> We do not parse the internal monologue of a single generation to extract "discarded attempts" as separate data points. Doing so would require a model-specific parser to distinguish between "exploration thoughts" and "final decisions," which would violate the **model-agnostic** design of our framework. However, we agree that analyzing the *intra-sample* diversity of the thinking trace is a fascinating avenue for future work to understand how models simulate divergent thinking internally. We will ensure our evaluation method is described in greater detail in the revised manuscript to avoid any ambiguity regarding how reasoning models are handled.
>
> **Question 2: Evaluation on Varying Difficulty Levels**
> >*"Model performance often varies across problems of different difficulty levels—for instance, on harder tasks, models typically require multiple rounds of trial and error before arriving at a correct solution. Would it be beneficial to evaluate models separately on problems of varying difficulty?"*
>
> ### **Response:**
>
> We agree that stratifying evaluation by problem difficulty would provide a higher-resolution view of model creativity. However, we did not include this stratification in the current paper for two primary reasons: the inherent subjectivity of "difficulty" in creative tasks and our focus on a domain-general framework.
>
> **1. The Challenge of Defining "Difficulty" in Creativity**
>
> In reasoning benchmarks like MMLU-PRO or code generation, difficulty is often objective (e.g., number of logical steps required). In open-ended creativity (e.g., BookMIA narrative generation), difficulty is highly subjective and context-dependent. Creating a "difficulty" label for our datasets (MacGyver, HypoGen, BookMIA) would require either (a) extensive human annotation, which limits scalability and contradicts our goal of **automated** evaluation, or (b) using an LLM-based difficulty scorer, which introduces a confounding variable (e.g., is the task actually hard, or does the judge model just lack knowledge in that domain?).
>
> **2. Domain-Generality vs. Task-Specific Heuristics**
>
> Our framework is designed to be **domain-agnostic**. "Difficulty" metrics are often tightly coupled to specific domains (e.g., "constraint count" for MacGyver vs. "plot complexity" for BookMIA). Integrating these would have required designing bespoke difficulty heuristics for every new domain, reducing the framework's generalizability.
>
> Nevertheless, this is a promising direction for future research. Subsequent work could explore automated difficulty proxies to distinguish model capability from prompt difficulty. Investigating how model performance degrades or adapts under increasing task constraints - consistent with theories on task complexity and the impact of restrictions on creative output (Amabile, 1996) - would provide a deeper understanding of the "creativity-difficulty" relationship.

---

> > ### Author Response · Authors · 2025-12-03
> > **Response to Reviewer GDEb [2/3]**
> >
> > **Question 3: Broader Thinking Models & Domain Correlations**
> > >*"Different “thinking” models often exhibit substantial performance differences. Could the paper include results from a broader set of thinking models? Additionally, is it possible to correlate these models’ creativity scores with their capabilities in domains like math or code?"*
> >
> > ### **Response:**
> >
> > We appreciate this suggestion and agree that a broader comparison would be valuable. However, a strict technical constraint limits our ability to benchmark proprietary closed-source reasoning models (such as Gemini 3 Pro or OpenAI’s GPT-5). Our divergent creativity metric, **Semantic Entropy**, relies on aggregating the **log-probabilities (logits)** of the generated tokens to quantify semantic uncertainty (see Eq. 1 and 3 in the paper). Currently, most commercial APIs do not provide access to these output logits - particularly for reasoning-heavy or "thinking" models - making it impossible to compute the metric for them. Consequently, we prioritized state-of-the-art open-weights models (like DeepSeek-R1-Distill and Qwen3 32B-Thinking), where we have full access to the probability distributions required for our evaluation.
> >
> > Regarding correlations with Math and Code capabilities: **correlating creativity scores with Math/Code benchmarks is both possible and highly insightful.** In fact, our results already offer strong preliminary evidence for this link:
> >
> > *   **Convergent Creativity & Math/Code:** We observed that models known for strong reasoning and coding performance (e.g., Qwen3 32B Thinking) achieved the highest scores in our **Convergent Creativity** metrics (Task Fulfillment, Feasibility). This supports the hypothesis that the logical structuring required for code/math transfers well to the constraint-satisfaction aspects of creative problem-solving.
> > *   **Divergent Creativity:** These same "math-heavy" models sometimes showed plateaued or lower Divergent Creativity (Semantic Entropy) in our tests (Section 6.1). This suggests that training heavily on convergent tasks (like math problems with a single correct answer) may potentially bias models toward a single "optimal" path, suppressing the exploration of the solution space.
> >
> > We believe a formal study correlating our Semantic Entropy scores against benchmarks like GSM8K (Math) or HumanEval (Code) would be a valuable addition to the literature to confirm if “reasoning" training trades off against "divergent" exploration.
> >
> > **Question 4: Independence of Divergent/Convergent Mechanisms**
> > >*"The paper states that divergent and convergent creativity stem from distinct mechanisms and can be optimized independently. However, does this independence depend on the problem type? For problems with very few viable solutions, achieving a high divergent creativity score may be inherently difficult—and attempting to artificially boost divergent creativity might compromise convergent performance. "*
> >
> > ### **Response:**
> >
> > We thank the reviewer for their inquiry, and agree that for a subset of tasks with extremely tight constraints or narrow solution manifolds, the number of valid diverse solutions is inherently low. In such cases, high divergence often implies error, leading to a negative correlation with convergent performance.
> >
> > However, our claim of independence is grounded in the **open-ended tasks** characteristic of our benchmark (MacGyver, HypoGen, BookMIA), where the solution space remains broad despite the constraints.
> >
> > **Clarification on "Boosting" vs. Measuring:**
> >
> > It is important to clarify that our evaluation does not attempt to "artificially boost" creativity (e.g., by injecting noise or forcing high-temperature randomness that would degrade coherence). Instead, we **measure** the model's inherent capacity for diversity by sampling responses to standard solution prompts.
> >
> > Our findings (Appendix G) show non-significant correlation between Semantic Entropy (Divergent) and Task Fulfillment (Convergent) across our datasets. This indicates that:
> >
> > 1.  **Joint Optimization is Possible:** High-performing models (like GPT-4o) demonstrate that it is possible to expand the "search radius" of valid ideas (high Divergence) without sacrificing the logical constraints required for the solution (high Convergence).
> > 2.  **Mechanistic Distinction:** The lack of a strong trade-off suggests that the capability to generate diverse candidates and the capability to verify/refine them are not strictly antagonistic. A model can theoretically be improved in its ability to explore the solution space without compromising its ability to adhere to constraints, provided the task allows for multiple valid approaches.

---

> > > ### Author Response · Authors · 2025-12-03
> > > **Response to Reviewer GDEb [3/3]**
> > >
> > > **Question 5: Adaptation as Reward Signal**
> > > >*"Could the proposed method be adapted in the future as a reward signal to enhance model diversity during training or inference?"*
> > >
> > > ### **Response:**
> > >
> > > We believe this is a promising avenue for future research, particularly for developing "Creativity-Aware" alignment strategies - e.g. RLAIF (Lee et al., 2023). Current alignment methods often penalize variance to ensure safety, potentially leading to mode collapse. By incorporating Semantic Entropy as an intrinsic reward signal, we could train models to maximize semantic diversity, while simultaneously using our Multi-Agent Judge to enforce task fulfillment. This would create a multi-objective optimization landscape that encourages exploration without sacrificing quality.
> > >
> > > Furthermore, this framework could guide inference-time strategies, such as creative tree search algorithms. A decoding system could use our metrics to detect when a model is repeating semantically identical ideas and force it to explore new conceptual clusters. While computing these metrics online is computationally intensive, future work could leverage offline scoring to create datasets for reward modeling or distillation, making scalable training for creativity feasible.
> > >
> > > ### **References**
> > >
> > > [1] Amabile, T.M. (1996). Creativity In Context: Update To The Social Psychology Of Creativity (1st ed.). Routledge. https://doi.org/10.4324/9780429501234
> > >
> > > [2] Lee, H., Phatale, S., Mansoor, H., Mesnard, T., Ferret, J., Lu, K., Bishop, C., Hall, E., Carbune, V., Rastogi, A., & Prakash, S. (2023, September 1). *RLAIF vs. RLHF: Scaling Reinforcement Learning from Human Feedback with AI Feedback*. arXiv.org. https://arxiv.org/abs/2309.00267

---

### Official Review · Reviewer_xSn2 · 2025-10-29

**Soundness:** 2
**Presentation:** 3
**Contribution:** 2
**Rating:** 4
**Confidence:** 3

**Summary:**

This paper presents an LLM evaluation framework for creative tasks. Following previous cognitive literatures, this paper advocates evaluating creativity from two perspectives: divergent creativity and convergent creativity. For the divergent creativity, this paper uses semantic entropy to measure the diversity and LLM responses. For the convergent creativity, this paper employs a multi-agent judge framework with a retrieval-based method for cost reduction. Experiments include testing the proposed framework on 3 creative datasets from different domains and several LLMs, which reveals the effectiveness of the proposed framework.

**Strengths:**

- This article focuses on the important problem of assessing LLM creativity and, grounded in cognitive science, advocates a systematic assessment of creativity in two aspects.
- Using semantic entropy, which is used to assess LLM hallucinations, to assess divergent creativity sounds interesting, as it seems to suggest that the two are deeply connected.

**Weaknesses:**

- A major weakness of this paper is the lack of technical innovation. The semantic entropy metric used to evaluate divergent creativity comes from previous work [1], while the multi-agent framework used to assess convergent creativity mainly comes from previous work [2] (with some improvements to reduce token consumption).
- The experimental part only involves three creativity-related benchmarks and lacks the verification of the proposed framework on creativity tasks in a wider range of domains.

Refs:

[1] Detecting hallucinations in large language models using semantic entropy

[2] Chateval: Towards better llm-based evaluators through multi-agent debate

**Questions:**

- Is using semantic entropy to assess divergent creativity affected by hallucinations? For example, when LLM responses contain hallucinations, will semantic entropy overestimate divergent creativity?
- Why is the accuracy of 'our framework: GPT-4o' better than ChatEval in Table 2? It seems that the main improvement of the proposed method is to save tokens.

---

> ### Author Response · Authors · 2025-12-03
> **Response to Reviewer xSn2 [1/2]**
>
> We thank the reviewer for the clear summary and constructive feedback.  Several of the reviewer’s concerns relate to the novelty of repurposing semantic entropy, the necessity of the multi-agent convergent judge, and the scope of domains evaluated. We address these points below, including **new human-annotated experiments** added in direct response to the reviewer’s feedback.
>
> ---
>
> **Weakness 1 (Part 1): Novelty of Semantic Entropy**
>
> > *"A major weakness of this paper is the lack of technical innovation. The semantic entropy metric used to evaluate divergent creativity comes from previous work [1], while the multi-agent framework used to assess convergent creativity mainly comes from previous work [2] (with some improvements to reduce token consumption)."*
>
> We thank the reviewer for raising this point and appreciate the opportunity to clarify the contribution of our use of Semantic Entropy (SE). The original SE formulation was demonstrated on single-answer QA tasks, where uncertainty reflects a model’s lack of confidence in the expected correct answer and was therefore introduced as a calibrated indicator of hallucination in the original work [1]. In contrast, the creative reasoning tasks we study are open-ended and admit multiple plausible continuations that can be simultaneously valid.
>
> **A central contribution of this paper is to establish and empirically validate a novel application of SE in this open-ended regime: using distributional spread to quantify divergent creative exploration rather than error.** In these settings, probability mass distributed across semantically distinct ideas does not indicate hallucination; instead, it reflects the model’s willingness to explore alternative conceptual directions – mirroring classical definitions of divergent creativity [2, 3]. We therefore hypothesize, and empirically confirm, that SE provides a meaningful signal of exploratory breadth in open-ended generative reasoning.
>
> To directly address the reviewer’s concern regarding theoretical and empirical grounding, we conducted **an additional human-annotated experiment** to evaluate whether SE aligns with human perceptions of divergent creativity.
>
> ---
> **Experimental Setup.**
>
> - We selected 50 MacGyver questions and generated model outputs from three LLMs (GPT-4o, Qwen3-32B-Think, Qwen3-32B). For each model, we computed SE using the same step-wise sampling protocol as in the benchmark, and additionally computed cosine similarity, Self-BLEU, and Distinct-n.
>
> **Human Annotation.**
>
> - Three independent raters evaluated the same model outputs. For each question, annotators performed pairwise comparisons (three per question), selecting the model showing greater *divergent creativity* – defined as broader conceptual directions, more varied strategies, and higher-level idea diversity. Majority vote produced a single human “gold” ranking of the LLMs for each question.
>
> ---
> **Results.**
>
> We evaluated each metric by computing Cohen’s κ agreement with the human-derived rankings.
>
> | Metric | κ |
> | --- | --- |
> | **Semantic Entropy (ours)** | **0.56** |
> | Cosine similarity | 0.49 |
> | Self-BLEU | 0.35 |
> | Distinct-1 | 0.37 |
> | Distinct-2 | 0.34 |
>
> ---
> **Interpretation:** Across all evaluated metrics, **SE achieves the highest agreement with human judgments**, exceeding both cosine similarity and surface-level lexical diversity metrics. This provides direct empirical evidence that:
>
> 1. SE captures the semantic breadth humans associate with divergent creativity, and is not purely limited to measurement of hallucination/error.
> 2. SE is a stronger predictor of human-perceived conceptual diversity than established diversity metrics.
>
> ---
>
> **Implication for novelty:**
>
> These findings reinforce the core novelty claim: **SE, when applied in open-ended creative reasoning, reliably reflects human-judged divergent creativity and constitutes a meaningful, empirically grounded extension beyond its original use in hallucination detection.** The additional study demonstrates that adapting SE is not only conceptually motivated but also supported by human-aligned empirical validation, addressing the reviewer’s concern about limited theoretical basis and insufficient novelty.
>
> ---
> References:
>
> [1] Farquhar, S., Kossen, J., Kuhn, L. et al. Detecting hallucinations in large language models using semantic entropy. Nature 630, 625–630 (2024).
>
> [2] Guilford, J. P. (1956). The structure of intellect. Psychological Bulletin, 53(4), 267–293.
>
> [3] Torrance, E.P. (1966) Torrance Test of Creative Thinking. Directions Manual and Scoring Guide. Personnel Press, Lexington.

---

> ### Author Response · Authors · 2025-12-03
> **Response to Reviewer xSn2 [2/2]**
>
> **Weakness 1 (Part 2): Novelty of Retrieval-based multi-agent framework**
>
> > *"A major weakness of this paper is the lack of technical innovation. The semantic entropy metric used to evaluate divergent creativity comes from previous work [1], while the multi-agent framework used to assess convergent creativity mainly comes from previous work [2] (with some improvements to reduce token consumption)."*
>
> We thank the reviewer for raising this point and would like to clarify the contribution of our convergent-creativity judge. While our framework draws inspiration from multi-agent debate, its role here is not a minor adaptation of ChatEval but a necessary component for assessing convergent creativity, which depends on multiple criteria – feasibility, safety, relevance, scientific validity, coherence – that a single LLM judge cannot reliably or consistently attend to at once. This limitation is reflected empirically: single-judge CoT reaches 67.3% accuracy, whereas a multi-agent setup reaches 76.7% (see Table 2, Results), showing that distributing evaluative focus across agents produces substantially more comprehensive judgments.
>
> This demonstrates that multi-agent assessment is **structurally important**, not incidental: convergent creativity requires evaluating correctness from several angles, and dividing these responsibilities across specialised agents yields more consistent and human-aligned decisions.
>
> However, existing multi-agent evaluators – most notably ChatEval – are computationally expensive to run at scale. ChatEval requires appending the *entire debate history* to each turn of the discussion, causing the prompt to grow with every agent exchange. As shown in Appendix Table 15, this results in an average of **75,578 tokens per question**, making it tricky to scale to 300 multi-step creative problems.
>
> **Our retrieval-based judge directly addresses this bottleneck.** Instead of replaying the full discussion each round, our framework retrieves only the most relevant past agent statements, preserving the benefits of multi-agent deliberation while dramatically reducing context length. This design reduces average token cost by **over 63%** (from 75,578 to 27,554 tokens per question, see Table 15) **without sacrificing accuracy**: when evaluated under the same GPT-4o backbone, our judge matches or exceeds ChatEval's accuracy while being substantially cheaper.
>
> Thus, **our contribution serves as an enabling mechanism that makes multi-agent convergent-creativity evaluation much more practical at benchmark scale.**
>
> ---
>
> **Weakness 2: Domain Generality**
>
> >*"The experimental part only involves three creativity-related benchmarks and lacks the verification of the proposed framework on creativity tasks in a wider range of domains."*
>
>
> We thank the reviewer for pointing out the concern regarding the experimental scope and domain generality. To clarify, our claim of “domain-general” assessment does *not* imply coverage of all modalities or tasks, but rather refers specifically to the framework’s ability to operate **uniformly across qualitatively different creative domains without task-specific modifications**.
>
> In our work, the *same* SE formulation, clustering procedure, and convergent-creativity judge are applied to:
>
> - **MacGyver** – functional, tool-based problem solving
> - **HypoGen** – scientific hypothesis generation
> - **BookMIA** – narrative/plot ideation
>
> These domains differ substantially in structure, linguistic form, and what constitutes “creative value.” The fact that a single methodology transfers across all three without domain-specific prompts, templates, or model retraining is the operational meaning of “domain-general” in this paper. This stands in contrast to most existing creativity-evaluation efforts, which typically confine themselves to a **single domain** such as story generation, metaphor generation, or code creativity [1–4].
>
> We will clarify this scope in the revised manuscript.
>
> ---
>
> References:
>
> [1]: Junyi Ye, Jingyi Gu, Xinyun Zhao, Wenpeng Yin, and Guiling Wang. Assessing the creativity of llms in proposing novel solutions to mathematical problems, 2024.
>
> [2]: Matthew DeLorenzo, Vasudev Gohil, and Jeyavijayan Rajendran. Creativeval: Evaluating creativity of llm-based hardware code generation. 2024 IEEE LLM Aided Design Workshop (LAD), pp. 1–5, 2024.
>
> [3]: John D. Patterson Paul V. DiStefano and Roger E. Beaty. Automatic scoring of metaphor creativity with large language models. Creativity Research Journal, 0(0):1–15, 2024.
>
> [4]: Carlos Gomez-Rodriguez and Paul Williams. A confederacy of models: a comprehensive evaluation of LLMs on creative writing. In Houda Bouamor, Juan Pino, and Kalika Bali (eds.), Findings of the Association for Computational Linguistics: EMNLP 2023, pp. 14504–14528, Singapore, December 2023. Association for Computational Linguistics.

---

### Official Review · Reviewer_fLUv · 2025-10-31

**Soundness:** 2
**Presentation:** 2
**Contribution:** 2
**Rating:** 4
**Confidence:** 3

**Summary:**

This paper focuses on evaluating the creativity of LLM-generated text, arguing that existing methods for assessing creativity are often task-specific and lack scalability. To address this issue, this paper proposes using semantic entropy to describe the innovativeness and diversity of LLMs and designs an automated, domain-agnostic framework. Experiments on three open-ended domains demonstrate that the proposed metrics capture novelty, diversity, and task fulfilment.

**Strengths:**

1. Addresses an important and timely problem— automated evaluation of creativity in large language models —which lacks standardized methodology.
2. A conceptually clear method for decomposing creativity is proposed, which decomposes creativity into two aspects: divergence (semantic entropy) and convergence (multi-agent judgment).

**Weaknesses:**

1. Limited novelty, as both proposed components largely adapts existing ideas. For example, semantic entropy was originally designed for hallucination detection. Since it primarily describes "uncertainty," its reinterpretation of "divergent creativity" lacks both theoretical foundation and empirical depth, and is only reflected in its relevance. Meanwhile, the retrieval-based multi-agent judging framework follows ChatEval with efficiency-oriented modifications, not a conceptual innovation.
2. Lack of validation with human judgment. The paper does not assess whether semantic entropy is related to human evaluation of creativity, therefore its construct validity is not demonstrated.
3. Experimental scope is limited. Only text-based tasks are tested, without exploration of multimodal or cross-domain generalization.

**Questions:**

1. Correlation analysis is indirect: it focuses on the relationships between metrics rather than the consistency between humans and metrics.
2. Please refer to Weaknesses.

---

> ### Author Response · Authors · 2025-12-03
> **Response to Reviewer fLUv [1/3]**
>
> We thank the reviewer for the thoughtful and constructive assessment of our work. Several of the reviewer’s concerns focus on the novelty of our reinterpretation of Semantic Entropy (SE), the need for human validation, and the experimental scope. We address each of these points in detail below, including **new human-annotated experiments** added in direct response to the reviewer’s feedback.
>
> ---
>
> **Weakness 1 (Part 1): Novelty of Semantic Entropy**
>
> > *"Limited novelty, as both proposed components largely adapts existing ideas. For example, semantic entropy was originally designed for hallucination detection. Since it primarily describes "uncertainty," its reinterpretation of "divergent creativity" lacks both theoretical foundation and empirical depth, and is only reflected in its relevance. Meanwhile, the retrieval-based multi-agent judging framework follows ChatEval with efficiency-oriented modifications, not a conceptual innovation."*
>
> We thank the reviewer for raising this point and appreciate the opportunity to clarify the contribution of our use of Semantic Entropy (SE). The original SE formulation was demonstrated on single-answer QA tasks, where uncertainty reflects a model’s lack of confidence in the expected correct answer and was therefore introduced as a calibrated indicator of hallucination in the original work [1]. In contrast, the creative reasoning tasks we study are open-ended and admit multiple plausible continuations that can be simultaneously valid.
>
> **A central contribution of this paper is to establish and empirically validate a novel application of SE in this open-ended regime: using distributional spread to quantify divergent creative exploration rather than error.** In these settings, probability mass distributed across semantically distinct ideas does not necessarily indicate hallucination; instead, it may reflect the model’s willingness to explore alternative conceptual directions – mirroring classical definitions of divergent creativity [2, 3]. We therefore hypothesize, and empirically confirm, that SE provides a meaningful signal of exploratory breadth in open-ended generative reasoning.
>
> To directly address the reviewer’s concern regarding theoretical and empirical grounding, we conducted **an additional human-annotated experiment** to evaluate whether SE aligns with human perceptions of divergent creativity.
>
> ---
> **Experimental Setup.**
>
> - We selected 50 MacGyver questions and generated model outputs from three LLMs (GPT-4o, Qwen3-32B-Think, Qwen3-32B). For each model, we computed SE using the same step-wise sampling protocol as in the benchmark, and additionally computed cosine similarity, Self-BLEU, and Distinct-n.
>
> **Human Annotation.**
>
> - Three independent raters evaluated the same model outputs. For each question, annotators performed pairwise comparisons (three per question), selecting the model showing greater *divergent creativity* – defined as broader conceptual directions, more varied strategies, and higher-level idea diversity. Majority vote produced a single human “gold” ranking of the LLMs for each question.
>
> ---
> **Results.**
>
> We evaluated each metric by computing Cohen’s κ agreement with the human-derived rankings.
>
> | Metric | κ |
> | --- | --- |
> | **Semantic Entropy (ours)** | **0.56** |
> | Cosine similarity | 0.49 |
> | Self-BLEU | 0.35 |
> | Distinct-1 | 0.37 |
> | Distinct-2 | 0.34 |
>
> ---
> **Interpretation:** Across all evaluated metrics, **SE achieves the highest agreement with human judgments**, exceeding both cosine similarity and surface-level lexical diversity metrics. This provides direct empirical evidence that:
>
> 1. SE captures the semantic breadth humans associate with divergent creativity, and is not purely limited to measurement of hallucination/error.
> 2. SE is a stronger predictor of human-perceived conceptual diversity than established diversity metrics.
>
> ---
>
> **Implication for novelty:**
>
> These findings reinforce the core novelty claim: **SE, when applied in open-ended creative reasoning, reliably reflects human-judged divergent creativity and constitutes a meaningful, empirically grounded extension beyond its original use in hallucination detection.** The additional study demonstrates that adapting SE is not only conceptually motivated but also supported by human-aligned empirical validation, addressing the reviewer’s concern about limited theoretical basis and insufficient novelty.
>
> ---
> References:
>
> [1] Farquhar, S., Kossen, J., Kuhn, L. et al. Detecting hallucinations in large language models using semantic entropy. Nature 630, 625–630 (2024).
>
> [2] Guilford, J. P. (1956). The structure of intellect. Psychological Bulletin, 53(4), 267–293.
>
> [3] Torrance, E.P. (1966) Torrance Test of Creative Thinking. Directions Manual and Scoring Guide. Personnel Press, Lexington.

---

> ### Author Response · Authors · 2025-12-03
> **Response to Reviewer fLUv [2/3]**
>
> **Weakness 1 (Part 2): Novelty of Retrieval-based multi-agent framework**
>
> > *"Limited novelty, as both proposed components largely adapts existing ideas. For example, semantic entropy was originally designed for hallucination detection. Since it primarily describes "uncertainty," its reinterpretation of "divergent creativity" lacks both theoretical foundation and empirical depth, and is only reflected in its relevance. Meanwhile, the retrieval-based multi-agent judging framework follows ChatEval with efficiency-oriented modifications, not a conceptual innovation."*
>
> We thank the reviewer for raising this point and would like to clarify the contribution of our convergent-creativity judge. While our framework draws inspiration from multi-agent debate, its role here is not a minor adaptation of ChatEval but a necessary component for assessing convergent creativity, which depends on multiple criteria – feasibility, safety, relevance, scientific validity, coherence – that a single LLM judge cannot reliably or consistently attend to at once. This limitation is reflected empirically: single-judge CoT reaches 67.3% accuracy, whereas a multi-agent setup reaches 76.7% (see Table 2, Results), showing that distributing evaluative focus across agents produces substantially more comprehensive judgments.
>
> This demonstrates that multi-agent assessment is **structurally important**, not incidental: convergent creativity requires evaluating correctness from several angles, and dividing these responsibilities across specialised agents yields more consistent and human-aligned decisions.
>
> However, existing multi-agent evaluators – most notably ChatEval – are computationally expensive to run at scale. ChatEval requires appending the *entire debate history* to each turn of the discussion, causing the prompt to grow with every agent exchange. As shown in Appendix Table 15, this results in an average of **75,578 tokens per question**, making it tricky to scale to 300 multi-step creative problems.
>
> **Our retrieval-based judge directly addresses this bottleneck.** Instead of replaying the full discussion each round, our framework retrieves only the most relevant past agent statements, preserving the benefits of multi-agent deliberation while dramatically reducing context length. This design reduces average token cost by **over 63%** (from 75,578 to 27,554 tokens per question, see Table 15) **without sacrificing accuracy**: when evaluated under the same GPT-4o backbone, our judge matches or exceeds ChatEval's accuracy while being substantially cheaper.
>
> Thus, **our contribution serves as an enabling mechanism that makes multi-agent convergent-creativity evaluation much more practical at benchmark scale.**
>
> ---
>
> **Weakness 2 and Question 1: Human judgement Validation**
>
> > *"Lack of validation with human judgment. The paper does not assess whether semantic entropy is related to human evaluation of creativity, therefore its construct validity is not demonstrated."*
>
> > *"Correlation analysis is indirect: it focuses on the relationships between metrics rather than the consistency between humans and metrics."*
>
> We thank the reviewer for highlighting the need for human-grounded validation. To directly address this point, we conducted **an additional human-annotated experiment** (see Response to Weakness 1) that evaluates whether Semantic Entropy aligns with human perceptions of divergent creativity.
>
> Across 50 MacGyver questions and three LLMs, **SE achieved the highest agreement with human-derived creativity rankings (κ = 0.56)**, outperforming cosine similarity, Self-BLEU, and Distinct-n (κ = 0.34–0.49). This result demonstrates two key findings relevant to the reviewer’s concern:
>
> 1. **SE corresponds closely to human-perceived divergent creativity.**
>
>     Human annotators judged models as more or less creative based on conceptual breadth, variety of strategies, and originality. SE tracked these judgments more closely than all lexical or embedding-based baselines, showing that it captures the *semantic* diversity humans value – not just surface-form variability or noise.
>
> 2. **SE is not simply measuring hallucination or error.**
>
>     If SE were dominated by hallucination-related uncertainty (its role in the original QA setting), we would expect weak or negative correlation with human creativity rankings. Instead, SE shows the strongest *positive* agreement with human judgments, validating its reinterpretation within open-ended creative reasoning.
>
>
> **Thus, the additional human study provides the construct-validity evidence the reviewer asked for.** It confirms that *in open-ended generative tasks*, SE is a meaningful and human-aligned operationalisation of divergent creativity. We hope this strengthens the contribution of the paper and demonstrates that the reinterpretation of SE is not merely conceptual but empirically validated.

---

> ### Author Response · Authors · 2025-12-03
> **Response to Reviewer fLUv [3/3]**
>
> **Weakness 3: Experimental scope and domain generality**
>
> >*"Experimental scope is limited. Only text-based tasks are tested, without exploration of multimodal or cross-domain generalization."*
>
>
> We thank the reviewer for pointing out the concern regarding the experimental scope and domain generality. To clarify, our claim of “domain-general” assessment does *not* imply coverage of all modalities or tasks, but rather refers specifically to the framework’s ability to operate **uniformly across qualitatively different creative domains without task-specific modifications**.
>
> In our work, the *same* SE formulation, clustering procedure, and convergent-creativity judge are applied to:
>
> - **MacGyver** – functional, tool-based problem solving
> - **HypoGen** – scientific hypothesis generation
> - **BookMIA** – narrative/plot ideation
>
> These domains differ substantially in structure, linguistic form, and what constitutes “creative value.” The fact that a single methodology transfers across all three without domain-specific prompts, templates, or model retraining is the operational meaning of “domain-general” in this paper. This stands in contrast to most existing creativity-evaluation efforts, which typically confine themselves to a **single domain** such as story generation, metaphor generation, or code creativity [1–4].
>
> However, we fully agree that extending the framework to multimodal or additional creative modalities is an important direction. We will clarify this scope in the revised manuscript and explicitly position multimodal extension as future work.
>
> ---
>
> References:
>
> [1]: Junyi Ye, Jingyi Gu, Xinyun Zhao, Wenpeng Yin, and Guiling Wang. Assessing the creativity of llms in proposing novel solutions to mathematical problems, 2024. https://arxiv.org/abs/2410.18336.
>
> [2]: Matthew DeLorenzo, Vasudev Gohil, and Jeyavijayan Rajendran. Creativeval: Evaluating creativity of llm-based hardware code generation. 2024 IEEE LLM Aided Design Workshop (LAD), pp. 1–5, 2024. URL https://api.semanticscholar.org/CorpusID:269148855.
>
> [3]: John D. Patterson Paul V. DiStefano and Roger E. Beaty. Automatic scoring of metaphor creativity with large language models. Creativity Research Journal, 0(0):1–15, 2024. URL https://doi.org/10.1080/10400419.2024.2326343.
>
> [4]: Carlos Gomez-Rodriguez and Paul Williams. A confederacy of models: a comprehensive evaluation of LLMs on creative writing. In Houda Bouamor, Juan Pino, and Kalika Bali (eds.), Findings of the Association for Computational Linguistics: EMNLP 2023, pp. 14504–14528, Singapore, December 2023. Association for Computational Linguistics. URL https://aclanthology.org/2023.findings-emnlp.966/.

---

### Official Review · Reviewer_chJR · 2025-10-31

**Soundness:** 1
**Presentation:** 2
**Contribution:** 1
**Rating:** 2
**Confidence:** 4

**Summary:**

This paper proposes an automated, domain-agnostic framework for evaluating LLM creativity, which it bifurcates into two components based on cognitive science concepts:
1. Divergent Creativity: To measure novelty and diversity, the paper adapts Semantic Entropy (SE), a metric originally developed for hallucination detection, as a reference-free measure of generative variability.
2. Convergent Creativity: To measure task fulfillment, the paper introduces a retrieval-based multi-agent judge framework. This framework aims to improve the computational efficiency of existing discussion-based evaluators by using a retrieval mechanism to condense the context, claiming over 60% improved efficiency and human-level accuracy.

The framework is validated on three distinct datasets—MacGyver (problem-solving), HypoGen (research ideation), and BookMIA (creative writing). The paper's main findings are that (1) SE is a robust metric for divergent creativity, (2) the retrieval-based judge is efficient and accurate, (3) divergent creativity (SE) does not improve with model size or recency, while convergent creativity does, and (4) this suggests divergent and convergent abilities are distinct mechanisms in LLMs.

**Strengths:**

1. The paper tackles a highly significant and challenging problem: the scalable, automated, and domain-general evaluation of creativity in LLMs. This is a crucial area of research for advancing generative AI.
2. The authors validate their framework across three qualitatively different creative domains (problem-solving, scientific ideation, and creative writing), which strengthens the claim of domain-generality.
3. The paper is supported by extensive appendices that provide full prompts, implementation details, and additional ablations, which aids in reproducibility.

**Weaknesses:**

1. The paper's primary metric for divergent creativity, semantic entropy (SE), is a metric for hallucination/uncertainty. Using it to measure "creativity" is fundamentally flawed and invalidates all conclusions drawn from it.
2. The framework's design is contradictory. It claims to measure divergent thinking by sampling 10 steps, but then uses greedy decoding to select the path forward. This inherently evaluates a convergent, high-probability path while claiming to measure the opposite.
3. The validation for semantic entropy is unconvincing. Figure 5a is tautological (SE vs. class count), and the main validation relies on circular logic (SE vs. an LLM-judge for novelty).
4. The paper's key finding, such that divergent and convergent creativity are "decoupled", is almost certainly an artifact of this flawed metric. A more plausible interpretation of the data (Fig 17, 18) is:
 - Convergent scores (task-fulfillment) improve with model size. This is known.
 - Divergent scores (SE, i.e., uncertainty/hallucination) do not improve or even decrease with model size. This is also known, as scaling and alignment reduce hallucination.
- The paper has not discovered a "decoupling of creativity." It has simply re-demonstrated that task-fulfillment and hallucination are (thankfully) not correlated.
5. The "novel" multi-agent judge provides no clear benefit. As shown in Table 2, the framework using a weak judge (GPT-4o-mini) performs worse than a simple one-shot baseline. This proves the framework's complexity is unjustified and all performance gains come from the judge's (GPT-4o) backbone.
6. "Domain-General" Claims Undermined.
 - The core SE metric relies on an entailment model, and Table 1 shows that changing this model causes accuracy to swing from 78.1% to 47.2%, indicating high instability.
 - The correlations between SE and other diversity metrics are wildly inconsistent across datasets (Tables 8, 9, 10), further suggesting the metric is not stable or domain-general.

**Questions:**

* Also see the weaknesses above

1. The paper's entire premise hinges on equating SE with divergent creativity. Can the authors provide a stronger justification for why high semantic uncertainty, which often leads to incorrect or nonsensical outputs, is a reliable proxy for creative (novel yet valuable) ideas?
2. How can the authors reconcile using greedy search ("most likely candidate") to build the final solution while claiming the framework measures divergent creativity? Doesn't this design ensure you are only evaluating the least creative, most probable path?
3. Given that "Our framework" with GPT-4o-mini (55.3% acc) performs significantly worse than the "Baselines" (e.g., One-shot 64.7%) in Table 2, how can you claim the framework itself provides any value? Doesn't this data prove that all the performance comes from the GPT-4o judge, and the retrieval framework itself is ineffective?
4. Please address this counter-hypothesis: The "divergent" metric (SE) measures uncertainty/hallucination. The "convergent" metric measures task-fulfillment. And the data shows that scaling improves task-fulfillment while reducing uncertainty. Is it not more accurate to conclude that this work simply re-proven that alignment reduces hallucination, rather than discovering a "decoupling of creativity"?
5. How do the authors defend the "model-agnostic" and "generalizable" claims when Table 1 shows that the choice of entailment model causes a ~30% absolute (from 47.2% to 78.1%) swing in performance?

---

> ### Author Response · Authors · 2025-11-23
> **Response to Reviewer CHJR [1/6]**
>
> We thank the reviewer for the thoughtful and detailed feedback. Several of the weaknesses and questions raised relate to shared underlying concerns (e.g., the interpretation of SE and the design of the two-stage framework), so we address these points together where appropriate. We appreciate the opportunity to clarify these aspects and outline below the relevant conceptual, empirical, and methodological explanations, along with planned manuscript revisions.
>
> ---
>
> **Weakness 1 & Question 1: Suitability of Semantic Entropy**
> >*"The paper's primary metric for divergent creativity, semantic entropy (SE), is a metric for hallucination/uncertainty. Using it to measure "creativity" is fundamentally flawed and invalidates all conclusions drawn from it."*
>
> >*"The paper's entire premise hinges on equating SE with divergent creativity. Can the authors provide a stronger justification for why high semantic uncertainty, which often leads to incorrect or nonsensical outputs, is a reliable proxy for creative (novel yet valuable) ideas?"*
>
> ### **Response:**
>
> We thank the reviewer for raising this important conceptual point and appreciate the opportunity to clarify. The original Semantic Entropy (SE) paper [1] evaluates SE on extractive and factoid QA datasets (TriviaQA, SQuAD, BioASQ, NQ-Open, SVAMP), where the task is framed around producing a specific gold answer. In this regime, uncertainty naturally indicates that the model is unsure about the expected correct answer, making SE a proxy for hallucination (as noted in the original SE work).
>
> **Our setting is fundamentally different.** MacGyver, HypoGen, and BookMIA are open-ended generative tasks with no single correct continuation. Because multiple distinct continuations can be simultaneously valid, the same “uncertainty” may no longer correspond to hallucination. Instead, when the model assigns probability mass to several semantically different continuation pathways, SE reflects the model’s willingness to explore diverse directions within the solution space. This interpretation is consistent with classical creativity theory, where divergent creativity is defined as the ability to generate multiple, varied, and non-obvious approaches when a problem admits more than one plausible solution [2, 3]. Since our tasks fall precisely into this open-ended regime, grouping candidate continuations into semantic classes and computing SE over these classes provides a natural operationalisation of how broadly the model explores alternative directions in the solution space.
>
> **Formally articulating and validating this novel application of SE for open-ended settings is one of the main contributions of our work.** Under this regime, SE measures distributional breadth in open-ended ideation, rather than error. We acknowledge that the manuscript did not make this distinction explicit, and we will revise Section 3.2 to clearly differentiate SE’s role in single-answer QA versus open-ended creativity tasks, and to articulate why SE reflects divergent generative exploration under this regime.
>
> Furthermore, **hallucination in these tasks is captured by our proposed convergent creativity as failures on the task-fulfilment criteria.** This follows prior creativity measure work such as [4, 5, 6] where convergent creativity accounts for the validity of the generation. For example, hallucinated or low-quality outputs appear as infeasible or unsafe steps in MacGyver, scientifically inaccurate hypotheses in HypoGen, or incoherent plot transitions in BookMIA. These behaviours are evaluated explicitly by our convergent-creativity LLM judge, which assesses metrics such as feasibility, safety, scientific accuracy, relevance, coherence, and plot completion across the three datasets (Section 5).
>
> Crucially, **SE does not correlate with these convergent metrics.** As shown in Figure 7 (and extended in Figs. 15-16), the correlation between SE and overall convergent creativity is low across all models and domains. If SE primarily reflected hallucination or error-similar to single-answer QA, we would expect a strong negative correlation with task fulfilment. This empirical separation indicates that SE is not tracking hallucination in our open-ended settings, and supports our hypothesis that SE captures distributional breadth in generative reasoning rather than model incorrectness.
>
> We thank the reviewer for the opportunity to clarify. We will elaborate on this relationship in the revised manuscript by explicitly noting that hallucination is evaluated in the convergent component, and by highlighting the empirical lack of correlation between SE and task-fulfilment metrics.

---

> ### Author Response · Authors · 2025-11-23
> **Response to Reviewer CHJR [2/6]**
>
> **Weakness 2 & Question 2: Sampling and Greedy Decoding Design**
> >*"The framework's design is contradictory. It claims to measure divergent thinking by sampling 10 steps, but then uses greedy decoding to select the path forward. This inherently evaluates a convergent, high-probability path while claiming to measure the opposite."*
>
> >*"How can the authors reconcile using greedy search ("most likely candidate") to build the final solution while claiming the framework measures divergent creativity? Doesn't this design ensure you are only evaluating the least creative, most probable path?"*
>
> ### **Response:**
>
> We thank the reviewer for this thoughtful concern and acknowledge that, in principle, generating multiple full solution paths may reveal more divergence at the trajectory level.
>
> However, **our goal is not to enumerate all possible full solutions.** The framework is designed to measure the model’s divergent creative potential - that is, the range of plausible approaches the model considers - rather than the creativity of a single final output. Divergent creativity, as defined in classical creativity theory [2, 3], concerns the ability to explore multiple, varied, and non-obvious approaches to a problem. Capturing this breadth may not require expanding full trajectories; analysing the distribution of possible next-step continuations could already reflect the range of approaches the model is considering at that point in the reasoning process. This aligns with the intuition of existing LLM reasoning methods such as Tree-of-Thoughts [7], where examining local branching factors provides a practical approximation to the model’s exploration space.
>
> **Expanding multiple full solutions may be less computationally efficient**, as it would require decoding 10 full trajectories per task, increasing cost by an order of magnitude and introducing exponential branching. Our step-wise sampling approach therefore offers a principled balance between diversity and efficiency: sampling 10 continuations at each reasoning step provides a tractable and theoretically grounded approximation of the model’s exploration space without the computational overhead of full-trajectory expansion.
>
> Additionally, **we use greedy decoding as an appropriate decoding strategy for the convergent component of our framework**. Classical creativity theory defines convergent creativity as the ability to arrive at a single correct, effective, or optimal solution (i.e. narrowing down multiple possibilities into one correct answer) [2, 3]. Evaluating this ability requires examining the model’s best and most confident continuation.
>
> We will clarify this two-stage design in the manuscript( Section 5, Experimental Setup) - and we appreciate the opportunity to clarify the distinction between sampling for divergent creativity and greedy decoding for convergent creativity.

---

> ### Author Response · Authors · 2025-11-23
> **Response to Reviewer CHJR [3/6]**
>
> **Weakness 3: Semantic Entropy Validation**
> >*"The validation for semantic entropy is unconvincing. Figure 5a is tautological (SE vs. class count), and the main validation relies on circular logic (SE vs. an LLM-judge for novelty)."*
>
> ### **Response:**
>
> We thank the reviewer for this observation and agree that, in general, **a larger number of semantic clusters will lead to higher SE.** This is expected: SE increases when the model spreads probability mass across a broader set of semantically distinct continuations. To clarify, the purpose of Figure 5a is to illustrate this expected behaviour by showing that higher SE corresponds to greater category-level dispersion in the generated ideas—directly aligning with the Torrance Tests of Creative Thinking (TTCT) [3] metric of flexibility, which measures the number of different idea categories produced. This connection demonstrates that SE reflects the semantic diversity predicted by creativity theory and provides the intended intuition for how the metric captures divergent generative breadth.
>
> We also thank the reviewer for raising the related concern regarding Figure 5d, and would like to clarify that the observed correlation is not circular. **The novelty judge does not evaluate uncertainty or distributional spread; its purpose is to assess qualitative markers of novelty in a single solution.** More specifically, the prompt asks: “How inventively each answer utilises tools, even when some steps are ordinary. Focus on unexpected tool applications.” (Appendix Page 40, Section L). These criteria are not functions of the model’s probability distribution or uncertainty and therefore are not directly tied to SE. Furthermore, the two metrics operate at different levels: SE evaluates dispersion across a set of generated continuations, whereas the novelty judge evaluates the novelty of a single selected solution. As such, the two measures capture distinct aspects of generative behaviour and do not form a circular relationship.
>
> In addition, **the novelty judge is independently validated with human annotations.** As shown in Figure 11 (Appendix), human annotators’ novelty ratings exhibit a Spearman correlation of 0.8 with the LLM-based novelty judge, indicating that the judge captures human-perceived semantic novelty rather than distributional uncertainty.
>
> We will clarify this distinction in the revised manuscript (Section 5) and make explicit that the novelty judge provides an independent semantic signal, rather than reflecting SE.

---

> ### Author Response · Authors · 2025-11-23
> **Response to Reviewer CHJR [4/6]**
>
> **Weakness 4 & Question 4: Divergent vs. Convergent Trends**
> >*"The paper's key finding, such that divergent and convergent creativity are "decoupled", is almost certainly an artifact of this flawed metric. A more plausible interpretation of the data (Fig 17, 18) is:*
> >*•Convergent scores (task-fulfillment) improve with model size. This is known.*
> >*•Divergent scores (SE, i.e., uncertainty/hallucination) do not improve or even decrease with model size. This is also known, as scaling and alignment reduce hallucination.*
> >*•The paper has not discovered a "decoupling of creativity." It has simply re-demonstrated that task-fulfillment and hallucination are (thankfully) not correlated."*
>
> >*"Please address this counter-hypothesis: The "divergent" metric (SE) measures uncertainty/hallucination. The "convergent" metric measures task-fulfillment. And the data shows that scaling improves task-fulfillment while reducing uncertainty. Is it not more accurate to conclude that this work simply re-proven that alignment reduces hallucination, rather than discovering a "decoupling of creativity"?"*
>
> ### **Response:**
>
> We thank the reviewer for this suggested interpretation and the chance to clarify. We agree with the first point raised: **larger (as well as more recent and reasoning-enhanced) models achieve higher task-fulfilment scores**, and this is already one of our reported findings (Section 6.2, Figure 6). However, the subsequent hypothesis - that SE reflects uncertainty or hallucination, and that the observed pattern merely restates the fact that alignment reduces hallucination - does not hold in our setting. As clarified in our earlier response (refer to Weakness #1 response), **SE does not measure hallucination in open-ended tasks with no single correct continuation**; instead, it reflects the distributional breadth of semantically distinct approaches the model considers. Indeed, SE is empirically uncorrelated with convergent task-fulfilment across all models and domains (Figure 7, Appendix Figures 15–16), which would not be the case if SE were tracking hallucination - hallucination would predict a strong negative correlation. This empirical independence supports our premise that SE reflects divergent generative breadth rather than model error.
>
> We hope the clarification addresses both the reviewer’s stated weakness and the counter-hypothesis posed in the accompanying question.
>
> With this premise clarified, **the trends in Figures 17–18 do not simply re-demonstrate that task fulfilment and hallucination are uncorrelated.** Rather, they show that the two components we measure - divergent breadth and convergent task fulfilment - do not necessarily scale together in open-ended generative tasks. However, we acknowledge that our original phrasing (“decoupling of creativity”) may have been too strong, as more extensive investigation across additional creative domains would be needed to make such a general claim. We will revise the manuscript to present this observation more cautiously and in line with classical creativity theory [2, 3], which treats divergent and convergent thinking as distinct cognitive processes rather than quantities that must increase in tandem.
>
> ---
> **Weakness 5 & Question 3: Judge Framework Effectiveness**
> >*"The "novel" multi-agent judge provides no clear benefit. As shown in Table 2, the framework using a weak judge (GPT-4o-mini) performs worse than a simple one-shot baseline. This proves the framework's complexity is unjustified and all performance gains come from the judge's (GPT-4o) backbone."*
>
> >*"Given that "Our framework" with GPT-4o-mini (55.3% acc) performs significantly worse than the "Baselines" (e.g., One-shot 64.7%) in Table 2, how can you claim the framework itself provides any value? Doesn't this data prove that all the performance comes from the GPT-4o judge, and the retrieval framework itself is ineffective?"*
>
> ### **Response:**
>
> We thank the reviewer for raising this point and would like to clarify that **all baselines, including the one-shot, few-shot, CoT, and ChatEval judges, are evaluated using GPT-4o**, whereas only our framework variant labelled “GPT-4o-mini” uses the smaller mini model. The lower accuracy of this ablation therefore reflects the weaker backbone rather than the retrieval-based multi-agent design itself. When evaluated under the same judge backbone (GPT-4o), our framework outperforms the baselines while also reducing token cost by over 60% (Appendix D.3), demonstrating that its benefits do not stem solely from model strength. We also emphasise that **efficiency, achieving competitive accuracy at significantly reduced cost, is the primary contribution of our framework**. We will revise the manuscript to make this distinction clearer in Table 2.

---

> ### Author Response · Authors · 2025-11-23
> **Response to Reviewer CHJR [5/6]**
>
> **Weakness 6 & Question 5: Entailment Model and Generality**
> >*""Domain-General" Claims Undermined.*
> >*•The core SE metric relies on an entailment model, and Table 1 shows that changing this model causes accuracy to swing from 78.1% to 47.2%, indicating high instability.*
> >*•The correlations between SE and other diversity metrics are wildly inconsistent across datasets (Tables 8, 9, 10), further suggesting the metric is not stable or domain-general."*
>
> >*"How do the authors defend the "model-agnostic" and "generalizable" claims when Table 1 shows that the choice of entailment model causes a ~30% absolute (from 47.2% to 78.1%) swing in performance?"*
>
> ### **Response:**
>
> We thank the reviewer for this observation and are happy to clarify the role of the entailment model. **The accuracy values in Table 1 do not reflect the stability or validity of SE**; they measure only how well each entailment classifier groups semantically similar ideas on a small, human-annotated clustering set (see Appendix C.2). Variation across models arises mainly from false negatives caused by overly strict entailment behaviour - for example, treating two expressions of the same underlying idea as different due to superficial lexical or tool-word differences. These variations reflect limitations of the entailment classifier model rather than instability in SE. The entailment model is a replaceable module, and selecting a model suitable for the domain improves clustering quality but does not affect the underlying SE formulation or the framework’s contribution. We acknowledge that **choosing an appropriate entailment model is important for maximising semantic clustering quality, but this is separate from the methodological contribution of the framework itself.**
>
> Regarding the reviewer’s concern about domain-generality, we would like to clarify what this claim entails in our work. The framework is applied uniformly across three qualitatively different creative domains, MacGyver (functional problem-solving), HypoGen (scientific hypothesis generation), and BookMIA (narrative ideation), using the same SE formulation and semantic clustering procedure. **This cross-domain applicability is what we mean by “domain-general.”** It does not imply invariance to the choice of auxiliary entailment classifier, but rather that the methodology operates independently of domain-specific prompts, templates, or training objectives. This stands in contrast to most prior creativity-evaluation approaches, which are usually restricted to a single domain [8-11].
>
> Additionally, regarding the correlations between SE and other diversity metrics, we thank the reviewer for highlighting this point. **The variability in correlations between SE and other diversity metrics across Tables 8-10 is expected and does not indicate instability.** The compared metrics primarily capture surface-form or lexical diversity, whereas SE measures semantic dispersion across idea classes. These axes of diversity are inherently different, and classical creativity theory likewise treats semantic flexibility as distinct from fluency or surface-level variety. Because each dataset emphasises a different type of creative variability, surface metrics respond differently across domains, while SE consistently captures semantic breadth.
>
> In this sense, **SE provides an orthogonal measure of creativity rather than replicating existing diversity metrics.** If SE were highly correlated with token-level metrics across all domains, it would suggest redundancy rather than a meaningful semantic signal. We will clarify this interpretation in the manuscript (Section 6.1).
>
> ---
> We thank the reviewer again for the constructive comments. We will revise the manuscript to incorporate the clarifications provided. We hope these responses satisfactorily address the reviewer’s concerns and help strengthen the final version of the paper.

---

> ### Author Response · Authors · 2025-11-23
> **Response to Reviewer CHJR [6/6]**
>
> ### **References:**
>
> [1] Farquhar, S., Kossen, J., Kuhn, L. et al. Detecting hallucinations in large language models using semantic entropy. Nature 630, 625–630 (2024).
>
> [2] Guilford, J. P. (1956). The structure of intellect. Psychological Bulletin, 53(4), 267–293.
>
> [3] Torrance, E.P. (1966) Torrance Test of Creative Thinking. Directions Manual and Scoring Guide. Personnel Press, Lexington.
>
> [4] Lu, Y., Wang, D., Li, T., Jiang, D., & Khashabi, D. (2024). Benchmarking Language model Creativity: A case study on code generation. arXiv (Cornell University). https://doi.org/10.48550/arxiv.2407.09007
>
> [5] Kumar, H., Vincentius, J., Jordan, E., & Anderson, A. (2024). Human Creativity in the Age of LLMS: Randomized Experiments on divergent and convergent thinking. arXiv (Cornell University). https://doi.org/10.48550/arxiv.2410.03703
>
> [6] Arora, V., Thabane, A., Parpia, S. et al. Generative artificial intelligence models outperform students on divergent and convergent thinking assessments. Sci Rep 15, 36987 (2025). https://doi.org/10.1038/s41598-025-21398-4
>
> [7] Yao, S., Yu, D., Zhao, J., Shafran, I., Griffiths, T. L., Cao, Y., & Narasimhan, K. (2023, May 17). Tree of Thoughts: Deliberate Problem Solving with Large Language Models. ArXiv.org. https://doi.org/10.48550/arXiv.2305.10601
>
> [8]: Junyi Ye, Jingyi Gu, Xinyun Zhao, Wenpeng Yin, and Guiling Wang. Assessing the creativity of llms in proposing novel solutions to mathematical problems, 2024. https://arxiv.org/abs/2410.18336.
>
> [9]: Matthew DeLorenzo, Vasudev Gohil, and Jeyavijayan Rajendran. Creativeval: Evaluating creativity of llm-based hardware code generation. 2024 IEEE LLM Aided Design Workshop  (LAD), pp. 1–5, 2024. URL https://api.semanticscholar.org/CorpusID:269148855.
>
> [10]: John D. Patterson Paul V. DiStefano and Roger E. Beaty. Automatic scoring of metaphor creativity with large language models. Creativity Research Journal, 0(0):1–15, 2024. URL https://doi.org/10.1080/10400419.2024.2326343.
>
> [11]: Carlos Gomez-Rodriguez and Paul Williams. A confederacy of models: a comprehensive evaluation of LLMs on creative writing. In Houda Bouamor, Juan Pino, and Kalika Bali (eds.), Findings of the Association for Computational Linguistics: EMNLP 2023, pp. 14504–14528, Singapore, December 2023. Association for Computational Linguistics. URL https://aclanthology.org/2023.findings-emnlp.966/.

---

> ### Author Response · Authors · 2025-12-03
> **Additional Experiments (addresses Weakness 2 and 3)**
>
> Motivated by the reviewer’s concerns, we conducted an additional experiment–including human annotations–to further evaluate (i) the role of greedy decoding in our two-stage design (Weakness 2) and (ii) the validity of SE as a robust measure of divergent creativity (Weakness 3).
>
> ---
> **Experimental Setup:** To isolate the effects raised by the reviewer, we designed a controlled comparison on 50 MacGyver questions using three LLMs (GPT-4o, Qwen 3 32b Think & Non-Think). The experiment introduces two decoding conditions:
>
> **1. Condition A – Greedy (original framework).**
>
> At each reasoning step, we sample k = 10 continuations to compute SE. The next step is then selected via greedy decoding, following the original benchmark design. This produces one full solution path and one SE score per model per question.
>
> **2. Condition B – Multi-path decoding (5-path variant).**
>
> At the *first* reasoning step, we sample *k = 10* continuations as usual, but instead of choosing the greedy path, we **decode 5 full solution paths** – yielding 5 independent trajectories. For each trajectory, we compute SE across its steps and then average these five SE scores to obtain a single SE value for that model. *(This differs from exponential decoding; only first branching step is expanded.)*
>
> This produces two SE estimates per model per question – one from Condition A and one from Condition B – under matched sampling conditions.
>
> ---
> **Human Annotation:** To obtain a human-grounded reference for divergent creativity, we collected annotations from three independent human raters on the same 50 MacGyver questions and model outputs used in this experiment.
>
> For each question:
>
> • Each of the three LLMs produced five full solution paths (matching Condition B).
>
> • Annotators conducted pairwise comparisons between model outputs (3 comparisons per question), selecting the model that exhibited higher divergent creativity.
>
>
> Each annotator therefore produced a pairwise preference matrix per question. We aggregated these by majority vote to obtain a single human “gold” ranking over the three models for each question.
>
> We then evaluated the two decoding regimes (SE under Condition A vs. SE under Condition B), along with other popular diversity metrics (cosine similarity, Self-BLEU, Distinct-n), by computing Cohen’s Kappa agreement with the human-derived rankings.
>
> ---
> **Results:** Across the 50 annotated MacGyver questions, the agreement coefficients (Cohen’s Kappa) are:
> | Metric | κ |
> | --- | --- |
> | Semantic Entropy (ours, greedy decoding; Condition A) | **0.56** |
> | Semantic Entropy (5-sample decoding; Condition B) | 0.48 |
> | Cosine similarity  | 0.49 |
> | Self-BLEU | 0.35 |
> | Distinct-1 | 0.37 |
> | Distinct-2 | 0.34 |
>
> Two observations follow:
>
> 1. Greedy-decoded SE aligns best with human judgments, outperforming all alternative diversity metrics and the 5-sample decoding variant.
> 2. SE (both variants) is competitive or superior to surface-form diversity metrics, which show substantially weaker agreement.
>
> ---
> **How This Supports Weakness 2 (Sampling vs. Greedy Decoding).**
>
> The reviewer was concerned that greedy decoding would obscure or suppress divergent creativity, yet our results show the opposite: **SE under greedy decoding aligns best with human judgments (κ = 0.56), outperforming both the multi-trajectory variant (κ = 0.48) and all other diversity metrics**. This indicates that greedy decoding does not weaken the divergent signal extracted from step-wise sampling; instead, it preserves the form of divergent potential that humans find most reflective of actual creative breadth.
>
> A plausible explanation for why greedy decoding outperforms the multi-trajectory variant is that expanding multiple full trajectories may cause different models to “sample into” similar diversity levels, reducing discriminability. Greedy decoding, by constraining the forward path to the model’s most confident continuation, reveals the model’s *intrinsic* distribution over semantically distinct ideas – leading to clearer separation between models and stronger agreement with human annotations.
>
> Overall, this experiment provides concrete empirical evidence that greedy decoding does not contradict the divergent-creativity component of the framework. On the contrary, it appears to be the more faithful operationalisation of the divergent potential captured by step-wise semantic entropy.
>
> ---
> **SE as a robust measure of Divergent Creativity (Weakness 3).**
>
> Among all evaluated diversity metrics, SE under greedy decoding achieves the highest agreement with human judgments (κ = 0.56), outperforming cosine similarity, Self-BLEU, and Distinct-n. This demonstrates that SE captures the semantic breadth that humans associate with divergent creativity more faithfully than popular surface-form diversity metrics. The experiment therefore provides additional empirical validation that SE is a robust and human-aligned measure of divergent creativity in open-ended generative tasks.

---

### Author Response · Authors · 2025-12-03
**Summary of Responses to all Reviewers**

We thank all reviewers for their thoughtful evaluations and valuable suggestions. Your comments helped us strengthen the technical clarity, broaden the empirical validation, and refine the overall positioning of our contributions. In response, we conducted new experiments and clarified key methodological choices.


### Key concerns raised and responses:

**1. New experiment with human annotation to validate the performance of Semantic Entropy and the Sampling & Greedy Decoding design**

See *Additional Experiments* Response for Reviewer chJR

**2. Suitability of Semantic Entropy as a measure of Divergent Creativity**

See *Response to Weakness 1 & Question 1* for Reviewer chJR

**3. Novelty of Semantic Entropy and Multi-Agent Framework**

See *Response to Weakness 1 (Both Part 1 and 2)* for Reviewer fLUv

**4. Validation of Semantic Entropy**

See (1) *Response to Weakness 3* for Reviewer chJR & (2) *Response to Weakness 2 & Question 1* for Reviewer fLUv

**5. Intepretation of Divergent vs. Convergent Trends**

See *Response to Weakness 4 & Question 4* for Reviewer chJR

**6. Entailment Model, stability of Semantic Entropy & Domain Generality Claims**

See (1) *Response to Weakness 6 & Question 5* for Reviewer chJR and (2) *Response to Weakness 3* for Reviewer fLUv

We will update the manuscript accordingly to incorporate all clarifications, revisions, and new experimental results. We believe these additions and clarifications significantly enhance the clarity of our contributions, and we hope the responses and updates address the concerns raised and strengthen the paper.

---

### Note · Authors · 2026-01-06

I have read and agree with the venue's withdrawal policy on behalf of myself and my co-authors.